# Minimal twister sister-like self-cleaving ribozymes in the human genome revealed by deep mutational scanning

Zhe Zhang[1,2,3], Xu Hong[1,2], Peng Xiong[2,3,4], Junfeng Wang[5,6], Yaoqi Zhou[1,3,7]*, Jian Zhan[1,3,8]*

[1]Institute for Systems and Physical Biology, Shenzhen Bay Laboratory, Shenzhen, China; [2]University of Science and Technology of China, Hefei, China; [3]Institute for Biomedicine and Glycomics, Griffith University, Southport, Australia; [4]Suzhou Institute for Advanced Research, University of Science and Technology of China, Suzhou, China; [5]High Magnetic Field Laboratory, Key Laboratory of High Magnetic Field and Ion Beam Physical Biology, Hefei Institutes of Physical Science, Chinese Academy of Sciences, Hefei, China; [6]Institute of Physical Science and Information Technology, Anhui University, Hefei, China; [7]School of Information and Communication Technology, Griffith University, Southport, Australia; [8]Ribopeutic Inc, Guangzhou International Bio Island, Guangzhou, China

**\*For correspondence:**
yaoqi.zhou@griffith.edu.au (YZ);
zhanjian@szbl.ac.cn (JZ)

**Competing interest:** The authors declare that no competing interests exist.

## eLife Assessment

This **important** study uncovers a surprising link between two self-cleaving RNAs that belong to the same structural family. The evidence supporting the main conclusions is **convincing** and based on extensive biochemical and bioinformatic analysis. This research will be of broad interest to RNA molecular biologists and biochemists.

**Abstract** Despite their importance in a wide range of living organisms, self-cleaving ribozymes in the human genome are few and poorly studied. Here, we performed deep mutational scanning and covariance analysis of two previously proposed self-cleaving ribozymes (LINE-1 and OR4K15). We found that the regions essential for ribozyme activities are made of two short segments, with a total of 35 and 31 nucleotides only. The discovery makes them the simplest known self-cleaving ribozymes. Moreover, the essential regions are circular permutated with two nearly identical catalytic internal loops, supported by two stems of different lengths. These two self-cleaving ribozymes, which are shaped like lanterns, are similar to the catalytic regions of the twister sister ribozymes in terms of sequence and secondary structure. However, the nucleotides at the cleavage site have shown that mutational effects on two twister sister-like (TS-like) ribozymes are different from the twister sister ribozyme. The discovery of TS-like ribozymes reveals a ribozyme class with the simplest and, perhaps, the most primitive structure needed for self-cleavage.

## Introduction

Ribozymes (catalytic RNAs) are thought to have dominated early life forms (**Benner et al., 1989**) until they were gradually replaced by more effective and stable proteins. One of the few that remain active is self-cleaving ribozymes found in a wide range of living organisms (**Ferré-D'Amaré and Scott, 2010**), involved in rolling circle replication of RNA genomes (**Forster and Symons, 1987**; **Hutchins**

*et al., 1986*; *Prody et al., 1986*), biogenesis of mRNA and circRNA, gene regulation (*Cervera and de la Peña, 2020*; *de la Peña and García-Robles, 2010a*; *de la Peña and García-Robles, 2010a*; *Martick et al., 2008*; *Salehi-Ashtiani et al., 2006*), and co-transcriptional scission of retrotransposons (*Eickbush and Eickbush, 2010*; *Sánchez-Luque et al., 2011*).

So far, there are a few self-cleaving ribozymes found in the human genome. They include two hammerhead ribozymes (*de la Peña and García-Robles, 2010a*; *Perreault et al., 2011*), an hepatitis delta virus (HDV)-like cytoplasmic polyadenylation element-binding protein 3 (CPEB3) ribozyme within the transcript of a single-copy gene (*Salehi-Ashtiani et al., 2006*), and self-cleaving ribozymes associated with olfactory receptor *OR4K15* (olfactory receptor family 4 subfamily K member 15), insulin-like growth factor 1 receptor (*IGF1R*), and a *LINE-1* (Long Interspersed Nuclear Element-1) retrotransposon (*Salehi-Ashtiani et al., 2006*). All these ribozymes were discovered by using selection-based biochemical experiments (*Salehi-Ashtiani et al., 2006*). More recently, the hovlinc ribozyme (*Chen et al., 2021*) was identified by genome-wide screening against RNAs with the specific 5′-hydroxyl termini resulted from self-cleavage, followed by a combination of RNA structure prediction as well as deletion analysis and mutation-based validation for determining its secondary structure. It has been shown that single-nucleotide polymorphism in CPEB3 ribozyme was associated with an enhanced self-cleavage activity along with a poorer episodic memory (*Vogler et al., 2009*). Inhibition of the highly conserved CPEB3 ribozyme could strengthen hippocampal-dependent long-term memory (*Bendixsen et al., 2021*; *Chen et al., 2024*). However, little is known about the other human self-cleaving ribozymes.

Previous studies on self-cleaving ribozymes found in retrotransposons suggested their importance in the overall size, structure, and function of different genomes. Most self-cleaving ribozymes found in retrotransposons belong to three widespread ribozyme families: twister ribozyme variants in non-LTR retrotransposons of *Schistosoma mansoni* (*Liu et al., 2021*), HDV-like ribozymes in R2 elements (*Eickbush and Eickbush, 2010*) and L1Tc retrotransposon (*Sánchez-Luque et al., 2011*) and hammerhead ribozymes in short interspersed nuclear elements of Schistosomes (*Ferbeyre et al., 1998*), Penelope-like elements (*Cervera and De la Peña, 2014*; *Lünse et al., 2017*), and retrozymes (*Cervera et al., 2016*). Unlike the wide occurrence of hammerhead ribozymes in different types of retrotransposons, HDV-like ribozymes usually locate in the 5′ UTR region of the retrotransposons only, and were found in different species including *Drosophila melanogaster* (*Dawid and Rebbert, 1981*; *Roiha et al., 1981*), *Bombyx mori* (*Eickbush and Robins, 1985*), arthropods, nematodes, birds, and tunicates (*Ruminski et al., 2011*; *Webb et al., 2009*). These ribozymes are important for co-transcriptional processing of the full-length transcript and translation of the downstream open reading frame. This suggests that the LINE-1 ribozyme found in the human genome may also play an active role in co-transcriptional processing during evolution.

Typically, self-cleaving ribozymes are in non-coding regions of the genome. For example, the CPEB3 ribozyme is located inside the intron of the *CPEB3* gene (*Salehi-Ashtiani et al., 2006*), hammerhead ribozyme found in the 3′ UTR of *Clec2* genes (*Martick et al., 2008*), hammerhead and HDV motifs associated with retrotransposable elements (*Cervera et al., 2016*; *Ruminski et al., 2011*). However, the OR4K15 ribozyme is located at the reverse strand of the coding region of the olfactory receptor gene *OR4K15*, suggesting a potentially novel functional mechanism for this ribozyme.

Previously, we performed deep mutational scanning of a self-cleaving ribozyme (CPEB3) by using error-prone PCR to generate mutants in a large scale. The relative activities of these mutants can be obtained by counting the cleaved and uncleaved sequences of each mutant from high-throughput sequencing data. These activities can be utilized to analyze mutational covariation and infer accurate base-pairing structures by developing a computational tool called CODA (covariation-induced deviation of activity) (*Zhang et al., 2020*). Here, using the same technique, we located the regions essential for the catalytic activities (the functional regions) of LINE-1 and OR4K15 ribozymes in their original sequences, which were generated from the selection of randomly fragmented human genomic DNA (*Salehi-Ashtiani et al., 2006*). We showed that these two ribozymes shared essentially the same base-pairing structural elements but with a circular permutation. The structural elements, which are shaped like the Chinese lantern, are homologous to the catalytic cores of the more complex twister sister self-cleaving ribozymes in terms of sequence and secondary structure, suggesting a more primitive origin for twister sister ribozymes. However, the homology model of the twister sister-like (TS-like) ribozyme generated from the twister sister ribozyme may not be the true structure for its function due to the

lack of a stem-loop for its stabilization and two mismatches from the twister sister in the internal loops as well as different responses to mutations.

## Results

### Deep mutational scanning of LINE-1 and OR4K15 ribozymes

Our previous work indicates that deep mutational scanning can lead to covariation signals for highly accurate inference of base-pairing structures (*Zhang et al., 2020*). Here, the same technique was applied to the original LINE-1 ribozyme (146 nucleotides, LINE-1-ori) and OR4K15 ribozyme (140 nucleotides, OR4K15-ori). These original sequences were generated from a randomly fragmented human genomic DNA selection-based biochemical experiments (*Salehi-Ashtiani et al., 2006*). Thus, their functional and structural regions are unknown. It needs to be noted that there is no exact match for LINE-1-ori and OR4K15-ori in the human genome (*Figure 1A*). This should be related to mutations accumulated during the selection. In deep mutational scanning, high-throughput sequencing can directly measure the relative cleavage activity (RA) (see Materials and methods) of each variant in the mutant libraries of these two ribozymes according to the ratio of cleaved to uncleaved sequence reads of the mutant, relative to the same ratio of the wild-type sequence. The relative activities of single mutations at each position indicate the sensitivity of cleavage activity to the mutations in that position. *Figure 1B* shows the average RA for a given sequence position for all LINE-1-ori mutants with single mutations at the position. Mutations in most positions in LINE-1-ori did not lead to large changes in RA. In other words, these sequence positions are unlikely to contribute to the specific structure required for the ribozyme to be functional. The activity of the LINE-1-ori ribozyme is only sensitive to the mutations in two short segments (54–71 and 83–99) where the RA can be reduced to nearly zero. Thus, the two terminal ends and the central region between bases 72 and 82 are not that important for LINE-1 self-cleavage activity. In *Figure 1C*, a similar distribution of RA was observed in OR4K15-ori. Only sequence positions inside the two short segments (70–84 and 101–116) are sensitive to mutations, which suggests that only these regions are required to form the functional RNA structure of the OR4K15 self-cleaving ribozyme. Thus, we have identified the contiguous functional regions of LINE-1 ribozymes (54–99, LINE-1-rbz) and OR4K15 ribozymes (70–116, OR4K15-rbz).

The deep mutation data were further employed to search for the base pairs inside these two ribozymes by using CODA analysis. CODA employed support vector regression to establish an independent-mutation model and a naive Bayes classifier to separate bases paired from unpaired (*Zhang et al., 2020*). Moreover, incorporating Monte-Carlo simulated annealing with an energy model and a CODA scoring term (CODA+MC) could further improve the coverage of the regions under-sampled by deep mutations. *Figure 1D, E* highlights the base pairs within the region 54–99 of LINE-1-ori (LINE-1-rbz) and the region 70–116 of OR4K15-ori (OR4K15-rbz), respectively. The CODA result for LINE-1-ori shows four base pairs (A14U34, A16U32, G17C31, C18G30) in one possible stem region, and another two base pairs (A4U43, A6A41) in the second possible stem region. Due to the lower mutational coverage, the CODA result for OR4K15-ori shows fewer base pairs than LINE-1-ori. Only two base pairs (U1A47, U3A45) in one possible stem region, and one lone base pair (A13U34) have been observed. The low mutation coverage for OR4K15-ori was due to the mutational bias (*Cadwell and Joyce, 1992*; *Keohavong and Thilly, 1989*) of error-prone PCR (*Figure 1—figure supplements 1–4*).

A few but strong covariation signals from CODA analysis can be expanded by Monte-Carlo (MC) simulated annealing as demonstrated previously (*Zhang et al., 2020*). Several contiguous base-paired regions were detected (*Figure 1D, E*). LINE-1-rbz has two highly reliable stem regions with a length of 6 nt, including one non-Watson–Crick pair A6A41. For OR4K15-rbz, the functional region has two reliable stem regions with lengths of 5 nt and 4 nt, respectively, including one non-Watson–Crick pair G14U33. Most of the base pairs discovered by MC simulated annealing are non-AU pairs, whose covariations are difficult to capture by error-prone PCR because of mutational biases (*Cadwell and Joyce, 1992*; *Keohavong and Thilly, 1989*). In addition, these two stems are consistent with the single mutation profiles. For example, two single mutations G14A and U33C showed high RA, 1.55 and 1.82, respectively, suggesting the high possibility of a G14U33 pair at these positions, because these two separate mutations change the wobble GU pair to the Watson–Crick AU and GC pairs, respectively, and lead to a more stable structure and higher activity. If G14U33 is true, a natural

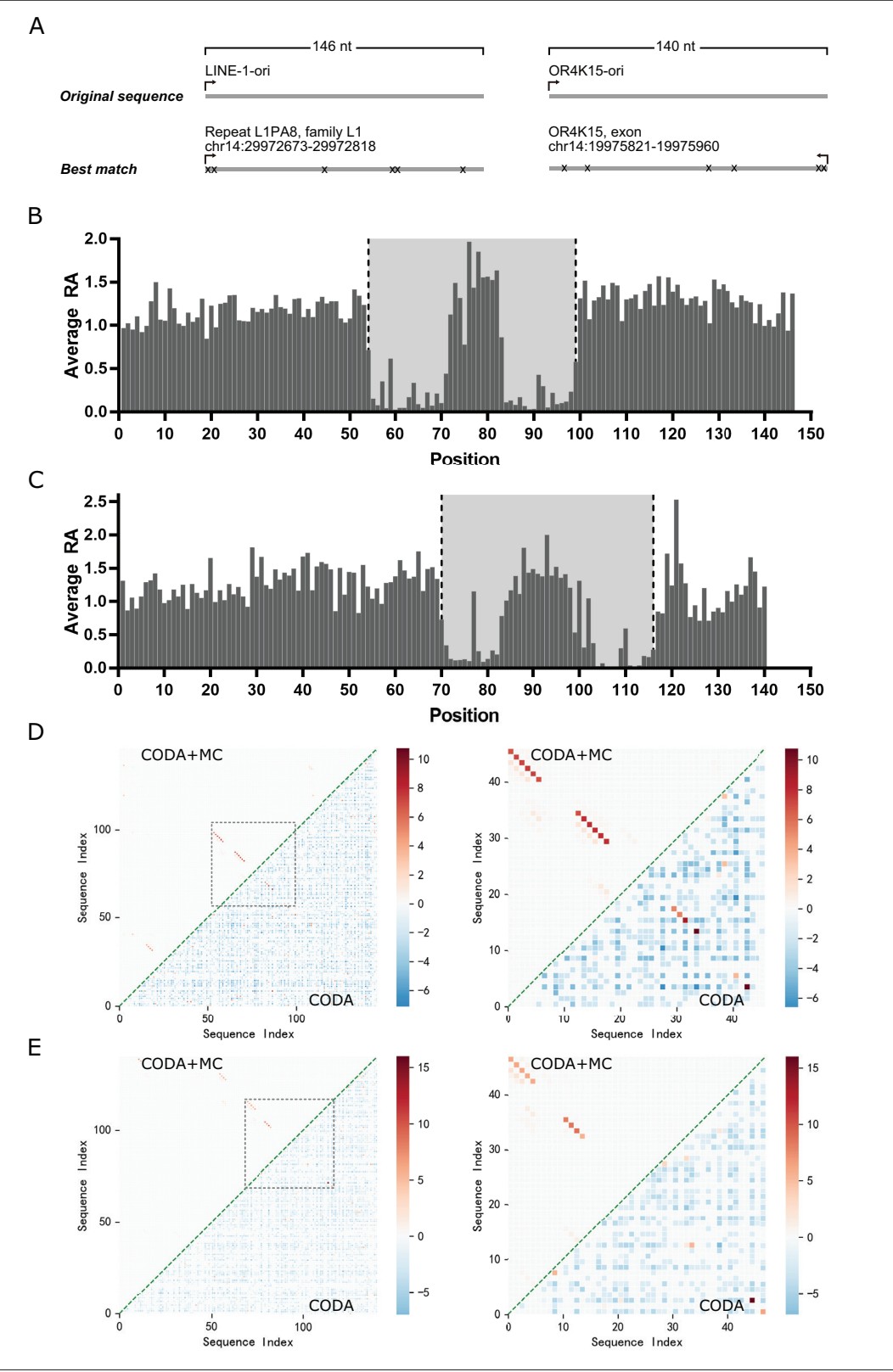

**Figure 1.** Deep mutational scanning results of the LINE-1 and OR4K15 ribozymes. (**A**) Genomic locations of the sequences with the highest similarity to the LINE-1 (left) and OR4K15 (right) ribozymes. (**B**) Average relative activity of mutations at each sequence position of the original LINE-1 ribozyme (LINE-1-ori). (**C**) Average relative activity of mutations at each sequence position of the original OR4K15 ribozyme (OR4K15-ori). (**D**) (Left) Base-pairing maps

*Figure 1 continued on next page*

*Figure 1 continued*

inferred from LINE-1-ori deep mutation data by covariation-induced deviation of activity (CODA) (lower triangle) and by CODA in combination with Monte-Carlo simulated annealing (CODA+MC) (upper triangle). (Right) Same as (Left) but for the contiguous functional region corresponds to 54–99 of LINE-1-ori (LINE-1-rbz). (**E**) (Left) Base-pairing maps inferred from OR4K15-ori deep mutation data by CODA (lower triangle) and by CODA+MC (upper triangle). (Right) Same as (Left) but for the contiguous functional region which corresponds to 70–116 of OR4K15-ori (OR4K15-rbz).

The online version of this article includes the following figure supplement(s) for figure 1:

**Figure supplement 1.** Mutation rates of two deep mutational scanning results at each nucleotide position of LINE-1 ribozyme.

**Figure supplement 2.** Mutation rates at each nucleotide position of OR4K15 ribozyme.

**Figure supplement 3.** Distribution of different mutation types in deep sequencing result of LINE-1 ribozyme.

**Figure supplement 4.** Distribution of different mutation types in the deep sequencing result of OR4K15 ribozyme.

**Figure supplement 5.** Single mutation-based average RA' and RA distribution.

**Figure supplement 6.** Experimental pipeline of two deep mutational scanning experiments.

extension of the stem region is G15C32 for another canonical Watson–Crick pair. This pairing region was missed by the CODA+MC result. There are some other base pairs that appeared in low probabilities after MC simulated annealing. They are considered as false positives because of low probabilities and lack of support from the deep mutational scanning results. The appearance of false positives is likely due to the imperfection of the experiment-based energy function employed in current MC simulated annealing.

## Further validation of base-pairing information by deep mutational scanning of LINE-1-rbz

To further improve the signals, we employed the contiguous functional segment (54–99, LINE-1-rbz) for the second round of deep mutational scanning (*Figure 1—figure supplement 5*). Performing the second round was necessary because the deep mutational scanning of the original LINE-1 ribozyme had a low 18.5% coverage of double mutations (*Supplementary file 1*). This low coverage was not sufficient for accurate inference of the secondary structure in the full coverage. This second round employed a chemically synthesized doped library with a doping rate of 6%, rather than a mutant library generated from error-prone PCR biased toward the sequence positions with A/T nucleotide (*Figure 1—figure supplements 1–4*, *Supplementary file 1*). In addition, to amplify the signals of cleaved RNAs after in vitro transcription of the mutant library of LINE-1-rbz, we selectively captured cleaved RNAs by employing RtcB ligase and a 5'-desbiotin, 3'-phosphate modified linker because they only react with the 5'-hydroxyl termini that exist only in cleaved RNAs (*Tanaka et al., 2011*). The captured active mutants were further enriched by the streptavidin-based selection after ligation. The technology improvement leads to less biased mutations in terms of the sequence positions (*Figure 1—figure supplements 1 and 2*) and mutation types (*Figure 1—figure supplements 3 and 4*). More importantly, it achieves 99.3% and 99.9% coverage of single and double mutations, respectively, within the contiguous functional segment (*Supplementary file 1*).

## Consistent secondary structure from deep mutational scanning of LINE-1-ori and LINE-1-rbz

We performed the CODA analysis (*Zhang et al., 2020*) based on the relative activities of 45,925 and 72,875 mutation variants (no more than 3 mutations) obtained for the original sequence and functional region of the LINE-1 ribozyme, respectively. CODA detects base pairs by locating the pairs with large covariation-induced deviations of the activity of a double mutant from an independent single-mutation model. *Figure 2A* shows the distribution of relative activity (RA', measured in the second round of mutational scanning) (see Materials and methods) of all single mutations for the LINE-1-rbz ribozyme, which is consistent with the distribution of RA in the functional region of LINE-1-ori ribozyme (*Figure 1A* and *Figure 1—figure supplement 1*). Moderate or high relative activity values (RA' > 0.5) at the two outer ends (18C, 30G, C46) of the stem regions suggest that these nucleotides

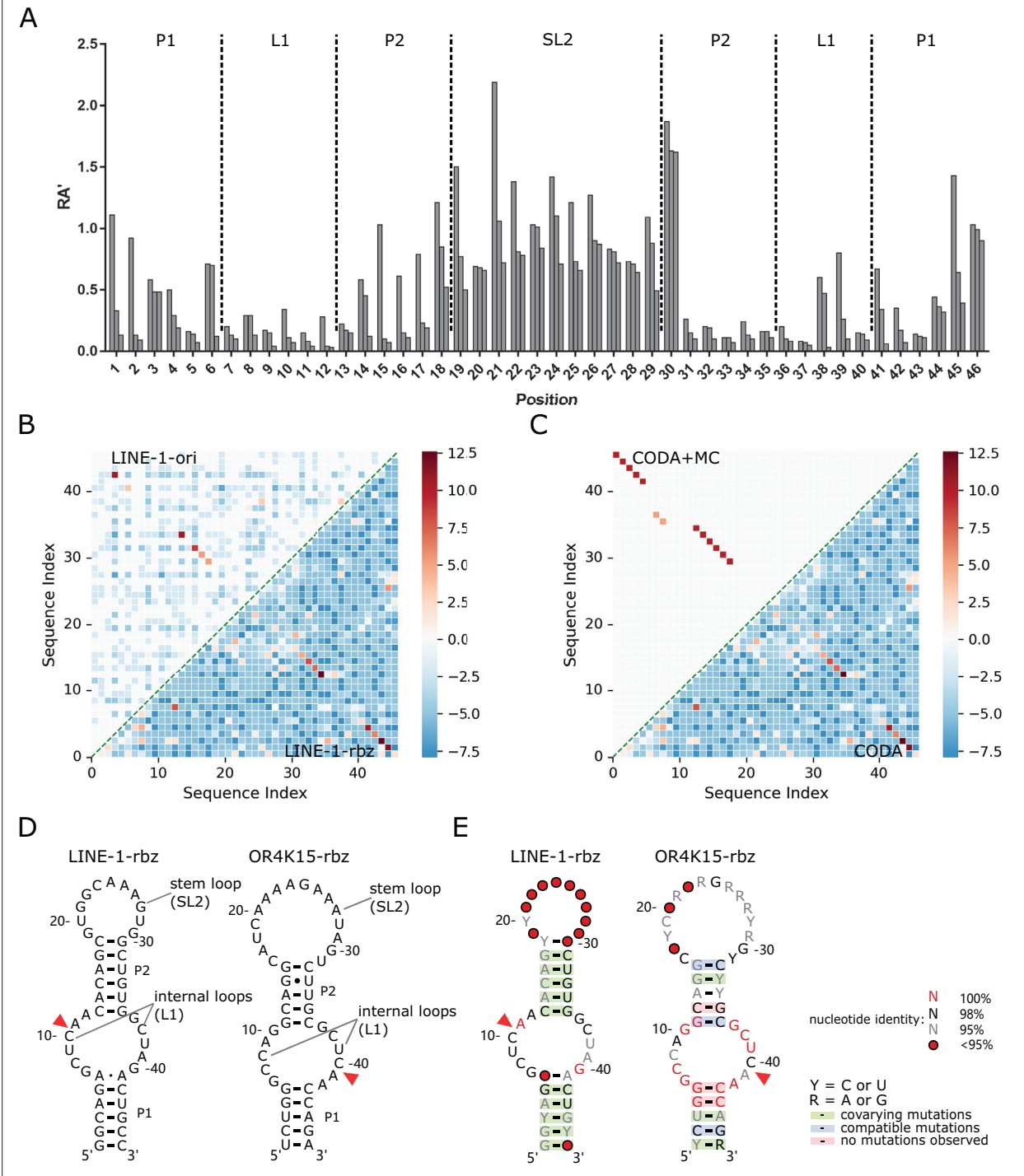

**Figure 2.** Secondary structures of the functional regions of the LINE-1 and OR4K15 ribozymes. (**A**) Relative activity of all mutations at each sequence position of the functional region of LINE-1 ribozyme (LINE-1-rbz). (**B**) Comparison between the base-pairing maps inferred from covariation-induced deviation of activity (CODA) analysis of deep mutational scanning of the functional region of LINE-1 ribozyme (LINE-1-rbz) (lower triangle) and that from the corresponding region (54–99 nt) in the deep mutational scanning of the original LINE-1 ribozyme (LINE-1-ori) (upper triangle). (**C**) Comparison between the base-pairing map inferred from LINE-1-rbz deep mutational data by CODA (lower triangle) and that after Monte-Carlo simulated annealing (CODA+MC, upper triangle). (**D**) Secondary structure model (left: LINE-1-rbz; right: OR4K15-rbz) inferred from the deep mutational scanning result. The red triangles indicate the cleavage sites of ribozymes. (**E**) Consensus sequence and secondary structure model (left: LINE-1-rbz; right: OR4K15-rbz) based on the alignment of functional mutants with RA' ≥ 0.5. The red triangles indicate the ribozyme cleavage sites. Positions with conservation of 95%, 98%, and 100% were marked with gray, black, and red nucleotides, respectively; positions in which nucleotide identity is less conserved are represented by circles. Green shading denotes predicted base pairs supported by covariation. R and Y denote purine and pyrimidine, respectively.

might not be necessary for the function of LINE-1-rbz, whereas RA' values at 1G were affected by the transcription efficiencies of different nucleotides downstream the T7 promoter.

*Figure 2B* shows that data from both the original sequence and functional region of the ribozyme reveal the existence of two stem regions. The latter clearly shows two stems with lengths of 6 nt and 5 nt, respectively, due to nearly 100% coverage of single and double mutations for the functional region. Moreover, it suggests a non-canonical pair 6A41A at the end of the first stem, although the signal is weak. Such a weak signal is expected because the non-canonical pairs are typically less stable than standard base pairings (*Lemieux and Major, 2002*). There are some additional weak signals in the LINE-1-rbz result. Most of these signals are for the base pairs at a few sequence distances apart ($|i - j| < 6$, local base pairs). We found that some of them are caused by a high relative activity (RA') of the double mutant and are not likely to be true positives because the corresponding single mutations are not very disruptive (RA' > 0.5). The consistency between base pairs inferred from deep mutational scanning of the original sequences and that of the identified functional regions confirmed the correct identification of functional regions for LINE-1 ribozyme.

Probabilistic CODA results were further refined by MC simulated annealing (CODA+MC). The resulting base pairs (*Figure 2C*, upper triangle) removed mostly local false positives. However, the non-canonical AA pair was removed as well, likely because the energy function does not account for non-canonical pairs. A new 18C30G pair added is a natural extension of the second stem. Two separate consecutive base pairs (7G37C, 8C36G) with relatively low signals were also added in the CODA+MC result. These two base pairs are likely false positives because they do not appear in all the models generated from MC simulated annealing with a low probability of 0.52. Taking all the information together leads to a confident secondary structure for the LINE-1-rbz ribozyme, which contains two stem regions (P1, P2), two internal loops, and a stem-loop (*Figure 2D*, left).

The consistent result between LINE-1-rbz and LINE-1-ori suggested that reliable ribozyme structures could be inferred by deep mutational scanning. This allowed us to use OR4K15-ori to directly infer the final inferred secondary structure for the functional region of OR4K15. The secondary structures for these two ribozymes (*Figure 2D*) are surprisingly similar to each other. Both ribozymes have two stems (P1, P2), two internal loops and a stem-loop region. Both stem-loop regions (SL2) are insensitive of ribozyme activity to mutations (*Figure 1B, C*). The internal loop regions (L1) of LINE-1-rbz and OR4K15-rbz are nearly identical except that C8 in OR4K15-rbz is replaced by U38 in LINE-1-rbz (*Figure 3A*).

## Consensus sequence of LINE-1-rbz and OR4K15-rbz

The consensus sequences (*Figure 2E*) for LINE-1-rbz and OR4K15-rbz were generated by R2R (*Weinberg and Breaker, 2011*) based on the multiple sequence alignment of 1394 and 621 mutants with RA $\geq$ 0.5 from deep mutational scanning. *Figure 2E* overlays the consensus sequences onto the secondary structure models from the CODA+MC analysis. The self-cleavage of the two ribozymes occurs between a conserved CA dinucleotide which is located inside the longer part of the internal loops (L1) linking P1 and P2. The internal loops are the regions with the highest conservation, which suggests their important roles in cleavage activity. Both stem regions (P1, P2) are not very conserved because of covariations. For example, in *Figure 2E* (right), 1U47A can be replaced by 1C47G, and 14G33U will form 14C33G after double mutations. Covariation of all CG pairs was missing in OR4K15-rbz due to mutational biases. The stem-loop regions show the lowest sequence identity, consistent with its insensitivity of catalytic activity to mutations in *Figure 1B, C*, providing additional support for the secondary structure models obtained here.

## The secondary structure of OR4K15-rbz is a circular permutation of LINE-1-rbz self-cleaving ribozyme

Removing the peripheral loop regions (*Figure 1B, C*) allows us to recognize that the secondary structure of OR4K15-rbz is a circular-permutated version of LINE-1-rbz. As shown in *Figure 3A*, after a permutation, OR4K15-rbz has two conserved internal loops with one loop identical to LINE-1-rbz, and the other loop differed from LINE-1-rbz by one base. The only mismatch U38C in L1 has the RA' of 0.6, suggesting that the mismatch is not disruptive to the functional structure of the ribozyme. As the simplest ribozymes reported so far, it is important to know if these two ribozymes share the same motifs with the known ribozyme families.

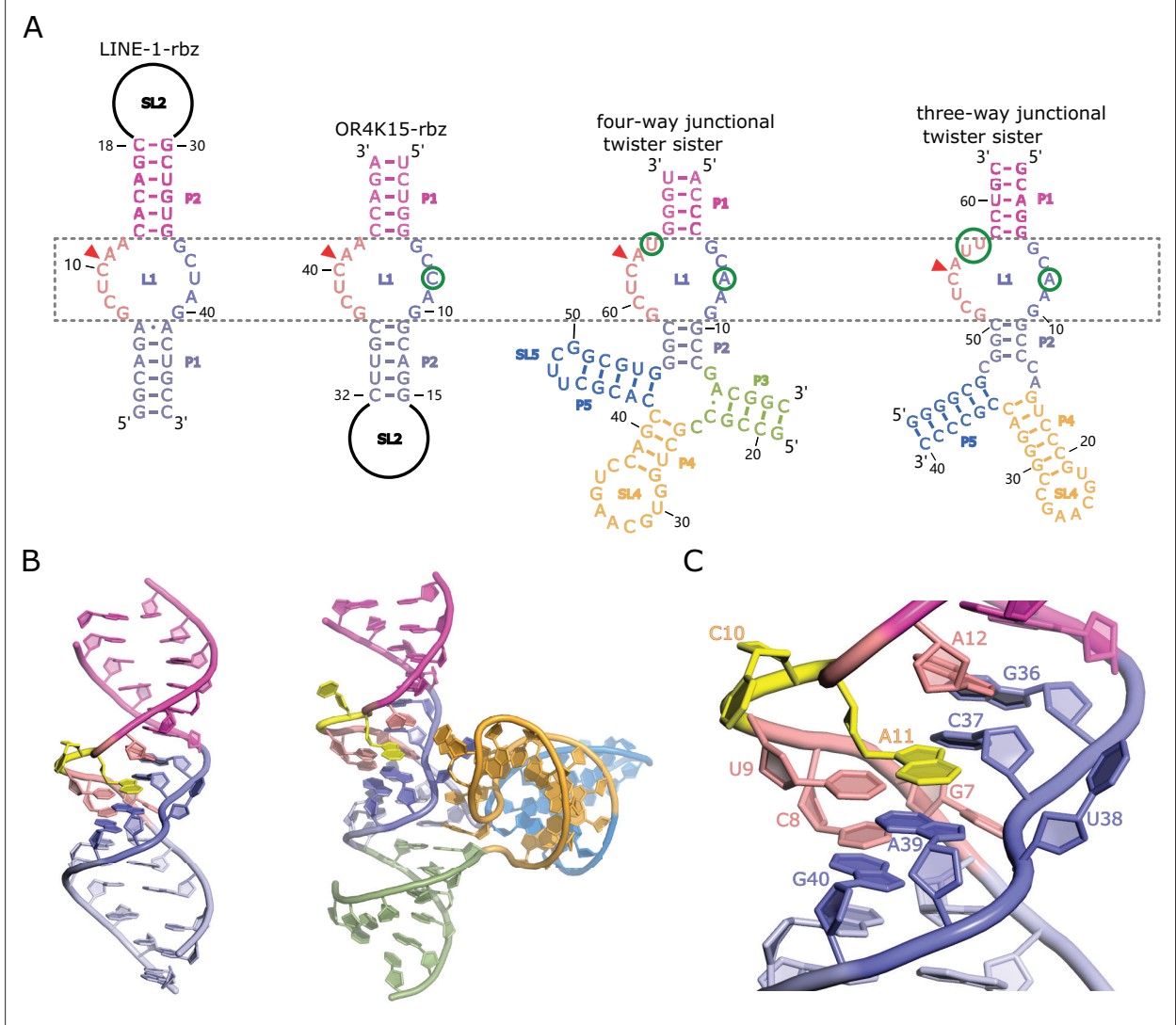

**Figure 3.** Homology-modeled structures of the bimolecular LINE-1-core and OR4K15-core. (**A**) Comparison between the secondary structure of LINE-1-rbz, OR4K15-rbz, four-way (PDB ID: 5Y87) and three-way junctional twister-sister ribozyme (PDB ID: 5T5A) reveals strikingly similar internal loops surrounding the cleavage sites (red triangles) with few nucleotide differences at each side of the internal loops L1. (**B**) A cartoon view of the homology-modeled structure of LINE-1-core (left) and four-way junctional twister-sister ribozyme (PDB ID: 5Y87) (right). (**C**) A detailed view of the catalytic core L1 of LINE-1-core in cartoon representation.

The online version of this article includes the following figure supplement(s) for figure 3:

**Figure supplement 1.** Structural patterns used for pattern-based similarity search.

**Figure supplement 2.** Homology-modeled structure of the bimolecular LINE-1-core.

**Figure supplement 3.** Base-pairing interactions in loop L1 of LINE-1-core that extend the stems P1 and P2.

**Figure supplement 4.** Homology modeled structure of the bimolecular OR4K15-core.

(**A**) A detailed view of the interactions in loop L1 of OR4K15-core. (**B**) Stacking interactions in loop L1 region. (**C**) The network of H-bonding interactions involving G37. (**D**) Intermolecular contacts at the C40-A41 cleavage site in the modeled structure. The scissile phosphate is colored magenta.

**Figure supplement 5.** Comparison between the secondary structure of LINE-1-rbz, OR4K15-rbz, and predicted secondary structure of the env-33 sequence from the multiple-sequence alignment of the twister sister ribozyme.

Here we used pattern-based similarity search (RNAbob, http://eddylab.org/software.html) to search the structural patterns against the known ribozyme families in Rfam database (*Figure 3—figure supplement 1*). The structural patterns of two ribozymes were derived from our deep mutational scanning results. We obtained three hits with identical motifs all from the twister sister ribozyme

class (*Supplementary file 2*). *Figure 3A* further shows the comparison with two previously published twister sister structures (*Liu et al., 2017*; *Zheng et al., 2017*). The two internal loops of LINE-1-rbz with 5 and 6 nucleotides, respectively, differ from the catalytic internal loops of the four-way junctional twister sister ribozyme only by one base in each loop (*Figure 3A*, *Zheng et al., 2017*). Given the identical lantern-shaped regions of the LINE-1-rbz and OR4K15-rbz ribozyme, we named them TS-like ribozymes. Thus, it is possible to use the structure of the twister sister ribozyme to build a homology model.

## Homology modeling of the TS-like ribozymes

To obtain the structure model of TS-like ribozymes, we used template-based homology modeling with the twister sister ribozyme structure as the template. *Figure 3B* shows the homology-modeled structure of bimolecular LINE-1-core (internal loops plus stems) built from the more identical four-way junctional twister sister ribozyme (PDB ID: 5Y87). The structure of the four-way junctional twister sister ribozyme revealed the internal loops L1 as the catalytic region involving a guanine–scissile phosphate interaction (G5–C62-A63), continuous stacking interactions, additional pairings and hydrated divalent Mg$^{2+}$ ions. *Figure 3B, C* and *Figure 3—figure supplement 2* show that the LINE-1-core model can be built as the twister sister ribozyme in the internal loops L1, with A11 at the cleavage site directed inwards. As shown in *Figure 3C* and *Figure 3—figure supplement 3*, stem P1 of LINE-1-rbz is extended by the Watson–Crick U9A39 as a part of the G7(U9A39) base triple and Watson-Crick

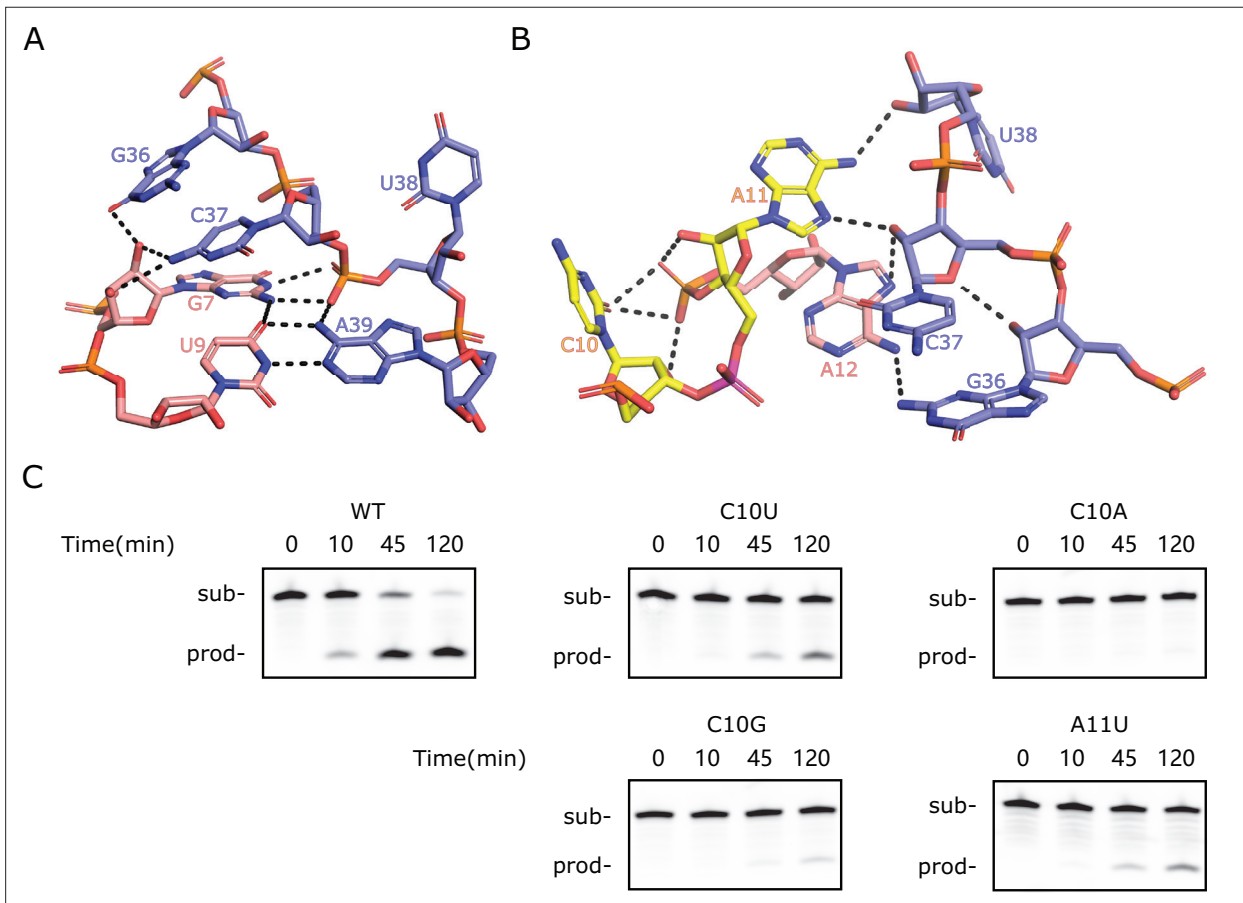

**Figure 4.** Special features of the homology-modeled LINE-1-core structure. (**A**) The network of H-bonding interactions involving G7. (**B**) Intermolecular contacts at the C10-A11 cleavage site in the modeled structure. The scissile phosphate is colored magenta. (**C**) PAGE analysis of the cleavage activity of wild-type (WT) and mutants C10U, C10G, C10A, and A11U.

The online version of this article includes the following source data for figure 4:

**Source data 1.** Original files for PAGE analysis displayed in *Figure 4C*.

**Source data 2.** Original files for PAGE analysis displayed in *Figure 4C*, with the relevant bands labeled.

C8G40. Meanwhile, stem P2 of LINE-1-rbz is extended through trans-non-canonical A12G36 and trans sugar edge-Hoogsteen A11C37. In addition to the triple base interaction in LINE-1-rbz, G7 is also H-bonded to both non-bridging O atoms of the phosphate connecting C37 and U38 (*Figure 4A*). This, along with the H-bond interaction between the *proR* O atom of the phosphate and A39 N6, generates a stable square array of H-bonds.

We also modeled the structure of OR4K15-core in the same way as LINE-1-core. As shown in *Figure 3—figure supplement 4A, B*, the model structure of L1 in OR4K15-core is similar to L1 in the LINE-1-core model, with stems P1 and P2 extended by additional base-pairing interactions in the tertiary structure of OR4K15-core. The base triple interaction of G37(A9U39) also exists in OR4K15-core, with the difference that G37 is H-bonded to A9 N6, rather than U39 (*Figure 3—figure supplement 4C*). Thus, the square array of H-bonds could not be observed as there was no direct interaction between A9 and the phosphate connecting C7 and C8. This modeling result suggests that the bond between A9 (A39 in LINE-1-core) and the phosphate may not be important, consistent with the fact that the replacement of the AU pair by a non-canonical UU pair did not disrupt cleavage activity (RA' = 0.8 for A39U in LINE-1-rbz).

Like the two twister sister ribozyme structures (*Liu et al., 2017*; *Zheng et al., 2017*), the nucleotide U38 in the LINE-1-core model (or C8 in the OR4K15-core model) is extruded from the helix (*Figure 3C*, *Figure 3—figure supplement 4A*). The corresponding nucleotides, A8 in three-way junctional twister sister ribozyme (*Liu et al., 2017*) and A7 in four-way junctional twister sister ribozyme (*Zheng et al., 2017*) are involved in the tertiary contact with the stem-loop SL4 (*Figure 3A*). Lacking the extra stem-loops in twister sister ribozyme, the key contacts for U38 of LINE-1-core (C8 in OR4K15-core) are the H-bond interaction with A11 N6 (A41 N6 in OR4K15-core), and the interactions between the phosphate and G7 (G37 in OR4K15-core). No involvement of the nucleobase part (U38 in LINE-1-core, C8 in OR4K15-core) in the modeled structures is compatible with the mismatches U38C and U38A found in the comparison, as changes of the nucleobase part did not have a big influence on the H-bonded interactions involving U38 (C8 in OR4K15-core) (*Figure 3A*). Thus, it is not entirely clear whether the model structure of TS-like ribozymes built from the twister sister will remain stable in the absence of any tertiary interactions with the stem-loop SL2 or it will deviate significantly from the twister sister.

## Differences in mutational responses in the L1s between the TS-like and the twister sister ribozymes

Considering the high similarity of the internal loops, we further investigated the mutational effects on the internal loop L1s. Most nucleotides in the L1 region of LINE-1-rbz are relatively conserved, as single mutations showed RA' < 0.2 according to deep mutational scanning results (*Figure 2A*). The conservation is consistent with the stacking and hydrogen-bonding interactions in the catalytic region. By comparison, mutations of C62 (C54 in three-way junctional twister sister ribozyme) at the cleavage site did not lead to a major change on the cleavage activity in previous studies (*Liu et al., 2017*; *Zheng et al., 2017*). This can be compared to the fact that single mutations of the corresponding nucleotide in LINE-1-rbz (C10) had relative activities as low as 0.07 in the deep mutational scanning result. To further confirm the above result, we performed cleavage assays on several mutations at the cleavage site of LINE-1-core. As shown in *Figure 4C*, mutations on C10 showed either partial (C10U, C10G) or complete loss (C10A) in cleavage activity. While the corresponding mutations in the four-way junctional twister sister ribozyme showed pronounced (C62U, C62A) or somewhat reduced (C62G) cleavage activity (*Liu et al., 2017*; *Zheng et al., 2017*). More interestingly, A11U in LINE-1-core showed partial cleavage activity, whereas the corresponding mutation A63U in the four-way junctional twister sister ribozyme showed complete loss of cleavage activity (*Liu et al., 2017*; *Zheng et al., 2017*). These different mutation effects indicate that LINE-1-core may not adopt exactly the same structure and catalytic mechanism as the twister sister ribozyme. In other words, TS-like ribozymes may not be simply a minimal version of twister sister ribozymes.

We further examined the interactions around the cleavage site C10-A11 in LINE-1-rbz (C40-A41 in OR4K15-core). The corresponding nucleotides in twister sister ribozymes (C62-A63 in four-way junctional twister sister ribozyme, C54-A55 in three-way junctional twister sister ribozyme) adopt either a splayed-apart or a base-stacked conformation, with the scissile phosphate H-bonded to the G5 (four-way junctional twister sister ribozyme) or C7 (three-way junctional twister sister ribozyme). In our modeled structures (*Figure 4B*, *Figure 3—figure supplement 4D*), the bases at the cleavage

site C10-A11 in LINE-1-core (C40-A41 in OR4K15-core) are splayed away, with A11 (A41 in OR4K15-core) directed inwards and C10 (C40 in OR4K15-core) directed outwards. In addition, H-bonded interactions between C10 (C40 in OR4K15-core) and the two nucleotides 3′ of C10 could be found (*Figure 4B*, *Figure 3—figure supplement 4D*). These structure differences may partially explain why mutations of C10 were detrimental to the catalysis in TS-like ribozymes. Moreover, we did not observe interactions between C10 (C40 in OR4K15-core) and G36/C37 (G6/C7 in OR4K15-core) although they were important to anchor the cytosine in twister sister ribozymes. This may be due to the replacement of the non-canonical pair G5U64 (G6U57 in three-way junctional twister sister ribozyme) by A12G36 (G6A42 in OR4K15-core) in TS-like ribozymes. Thus, the stability of the current model structure based on the template from the twister sister ribozyme is uncertain.

## Activity confirmation and biochemical analysis of LINE-1-core and OR4K15-core

To confirm the cleavage activity of the core region (without the stem-loop linker), we obtained the segments (54–71 for LINE-1, 101–116 for OR4K15) as the substrate strands with the cleavage site and the segments (83–99 for LINE-1, 70–84 for OR4K15) as the enzyme strands (*Figure 5A*), as they made up the LINE-1-core and OR4K15-core. When the substrate strand was mixed with the enzyme strand, most of the substrate strand was cleaved in the presence of $Mg^{2+}$ (*Figure 5B*). When $Mg^{2+}$ was not included in the reaction, no cleavage could be observed even after 24 h's incubation (*Figure 5— figure supplement 1*). This result confirms that the RNA cleavage in TS-like ribozymes is a catalytic reaction, accelerated by $Mg^{2+}$. Moreover, the result confirmed that the stem-loop SL2 regions in LINE-1-rbz and OR4K15-rbz (*Figure 2*) did not participate in the catalytic activity.

The core constructs of the two TS-like ribozymes contain 35 and 31 nucleotides, respectively. They are the simplest and the second simplest self-cleaving ribozymes reported so far. In all previously reported self-cleaving ribozyme families, the self-cleaving reaction occurs through an internal phosphoester transfer mechanism, in which the 2′-hydroxyl group of the −1 (relative to the cleavage site) nucleotide attacks the adjacent phosphorus resulting in the release of the 5′ oxygen of +1 (relative to the cleavage site) nucleotide (*Ferré-D'Amaré and Scott, 2010*; *Harris et al., 2015*; *Li et al., 2015*; *Roth et al., 2014*; *Weinberg et al., 2015*). We confirmed that both LINE-1-core and OR4K15-core employed the same cleavage mechanism because the analogous RNAs that lacked the 2′ oxygen atom in the −1 nucleotide (dC10 for LINE-1-core, dC40 for OR4K15-core) were unable to cleave (*Figure 5B*). We also investigated whether TS-like ribozymes can cleave when $Mg^{2+}$ was replaced by $Co(NH_3)_6^{3+}$ in the reaction. $Co(NH_3)_6^{3+}$ is isosteric with $Mg(H_2O)_6^{2+}$, but the divalent cations cannot directly participate in catalysis as the amino ligands cannot readily dissociate. In *Figure 5C and a* total loss of cleavage activity in $Co(NH_3)_6^{3+}$ in the absence of $Mg^{2+}$ indicates that the TS-like ribozymes require the binding of divalent cations not only for structure folding, but also for catalysis.

As shown in *Figure 5D*, we further examined the dependence of the metal ions on TS-like ribozymes' cleavage activity. At a concentration of 1 mM, ribozyme cleavage can be observed with $Mg^{2+}$, $Mn^{2+}$, $Co^{2+}$, and $Zn^{2+}$ but little or none with $Ca^{2+}$, $Cu^{2+}$, $Ba^{2+}$, $Ni^{2+}$, $Na^+$, $K^+$, $Li^+$, $Cs^+$, or $Rb^+$, indicating that direct participation of specific hydrated divalent metal ions is required for self-cleavage. More interestingly, we found that the TS-like ribozymes have an equivalent or even higher cleavage ratio with $Mn^{2+}$ than $Mg^{2+}$. For LINE-1-core, the cleaved fractions were ~57% for $Mg^{2+}$, ~74% for $Mn^{2+}$. For OR4K15-core, the cleaved fractions were ~9% for $Mg^{2+}$, ~79% for $Mn^{2+}$. We further characterized the cleavage rates of these two ribozymes under different concentrations of $Mg^{2+}$ and $Mn^{2+}$ by using a fluorescence resonance energy transfer-based method, as shown in *Figure 5—figure supplements 2 and 3*. We used the first-order rate constant ($k_{obs}$) to represent the efficiency of the cleavage reaction. The $k_{obs}$ of LINE-1-core under single-turnover condition was ~0.05 $min^{-1}$ when measured in 10 mM $MgCl_2$ and 100 mM KCl at pH 7.5 (*Figure 5—figure supplement 4*). Only a slightly lower value of $k_{obs}$ (~0.03 $min^{-1}$) was observed for LINE-1-rbz (*Figure 5—figure supplement 5*). This confirms that the stem-loop region SL2 does not contribute much to the cleavage activity of the TS-like ribozymes. Cleavage activities of the two TS-like ribozymes were highly dependent on the concentration of divalent metal ions. The steep increase in rate constants of two TS-like ribozymes plateaued at $Mg^{2+}$ concentrations above 100 mM, while for $Mn^{2+}$ it plateaued at concentrations of 10 mM. Consistent with the PAGE result, LINE-1-core showed higher cleavage rates than OR4K15-core when they were incubated with $Mg^{2+}$. However, the cleavage rates of OR4K15-core were only lower than LINE-1-core

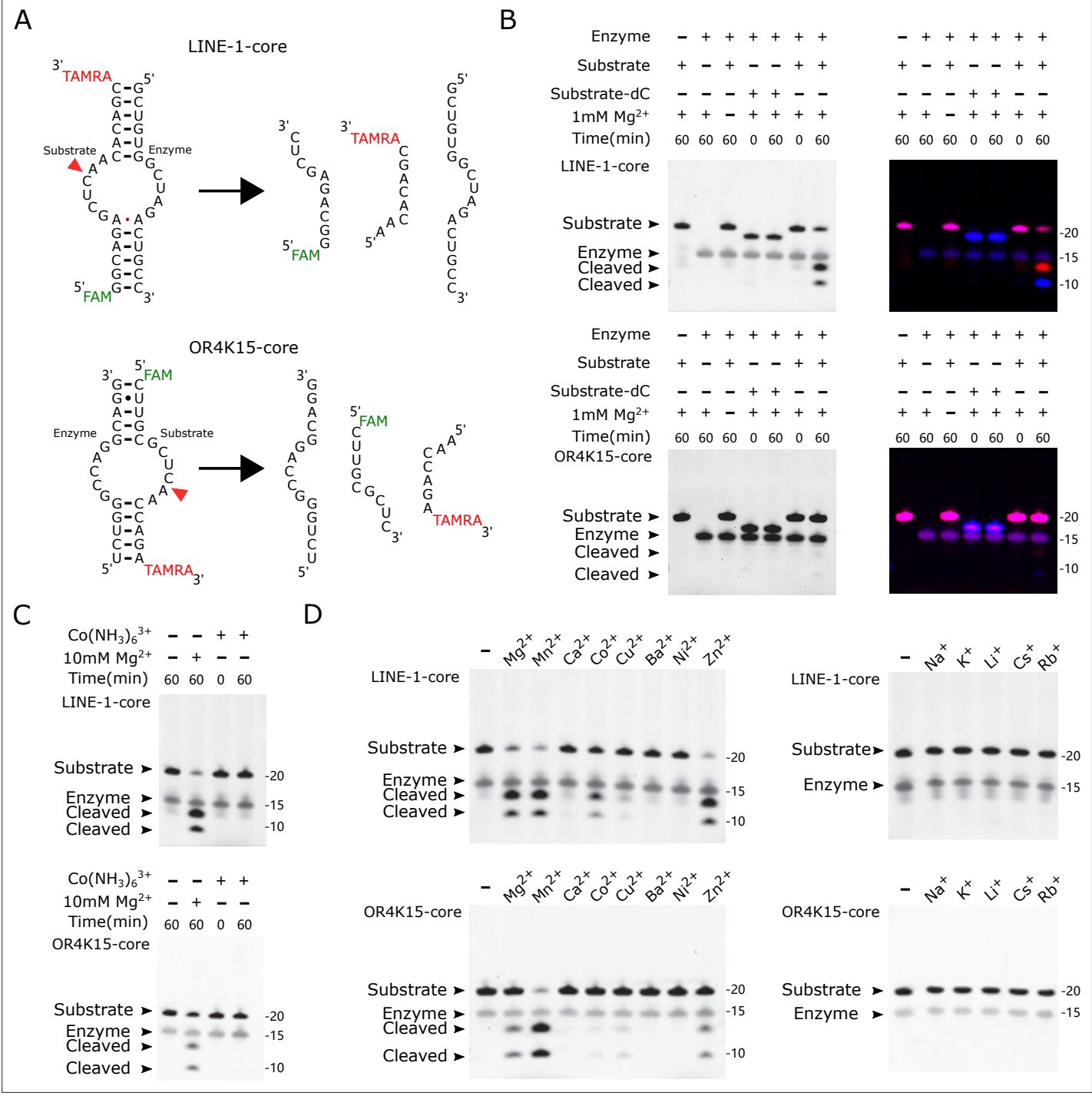

**Figure 5.** Biochemical assays of LINE-1-core and OR4K15-core. (**A**) (Top) The components of the bimolecular construct of LINE-1-core during cleavage, substrate strand (54–71) and enzyme strand (83–99). (Bottom) The components of the bimolecular construct of OR4K15-core during cleavage, the enzyme strand (70–84) and substrate strand (101–116). The red arrowhead indicates the cleavage site. (**B**) PAGE-based cleavage assays of the bimolecular construct with the all-RNA substrate and a substrate analog (dC) wherein a 2′-deoxycytosine is substituted for the cytosine ribonucleotide at position 10 or 40 of the substrate RNA. The substrate RNA was incubated with the enzyme RNA at 37°C for 1 hr in the presence (+) or absence (−) of 1 mM MgCl₂. The single-channel fluorescent images (left) were generated by using UV excitation (302 nm) and 590/110 nm emission on the ChemiDoc MP imaging system. The multi-channel fluorescent images (right) are overlays of two scans. They were generated from the ChemiDoc MP imaging system (Bio-Rad), Fluorescein (excitation: Epi-Blue 460–490 nm, emission: 532/28) for FAM, Cy3 (excitation: Epi-Green 520–545 nm, emission: 602/50) for TAMRA. (**C**) Cleavage assays in the absence (−) or presence of cobalt hexammine chloride [Co(NH₃)₆Cl₃] or MgCl₂ at 5 mM for 1 hr. (**D**) Cleavage

*Figure 5 continued on next page*

*Figure 5 continued*

assays at 37°C for 1 hr, in the absence (−) or presence (+) of various metal ions. Divalent metal ions (left) were used at a final concentration of 1 mM, and monovalent metal ions (right) were used at a final concentration of 1 M.

The online version of this article includes the following source data and figure supplement(s) for figure 5:

**Source data 1.** Original files for PAGE analysis displayed in *Figure 4C*.

**Source data 2.** Original files for PAGE analysis displayed in *Figure 4C*, with the relevant bands labeled.

**Figure supplement 1.** PAGE-based cleavage assay of LINE-1-core in a 24-hr range.

**Figure supplement 1—source data 1.** Original files for PAGE analysis displayed in *Figure 1*.

**Figure supplement 1—source data 2.** Original files for PAGE analysis displayed in *Figure 1*, with the relevant bands labeled.

**Figure supplement 2.** The dependence of two twister sister-like (TS-like) ribozymes' rate constants on $Mg^{2+}/Mn^{2+}$ concentration.

**Figure supplement 3.** A representative time course for the bimolecular ribozyme construct LINE-1-core under the conditions indicated as used for determining $k_{obs}$ values.

**Figure supplement 4.** The bimolecular ribozyme construct LINE-1-core with different E/S ratios (molar ratio between the enzyme strand and the substrate strand) for determining $k_{obs}$ values.

**Figure supplement 5.** Kinetic analysis using the single-stranded LINE-1-rbz and OR4K15-rbz.

**Figure supplement 5—source data 1.** Original files for PAGE analysis displayed in *Figure 5*.

**Figure supplement 5—source data 2.** Original files for PAGE analysis displayed in *Figure 5*, with the relevant bands labeled.

when the concentrations of $Mn^{2+}$ were less than 20 mM, but higher, otherwise. Thus, there may exist an important difference between OR4K15-core and LINE-1-core to explain this different bias toward ions. A detailed explanation for this difference may require high-resolution structure determination of the TS-like ribozymes.

## Homology search of two TS-like ribozymes

To locate close homologs of the two TS-like ribozymes, we performed cmsearch based on a covariance model (*Eddy and Durbin, 1994*) built on the sequence and secondary structural profiles. In the human genome, we got 1154 and 4 homolog sequences for LINE-1-rbz and OR4K15-rbz, respectively. For OR4K15-rbz, there was an exact match located at the reverse strand of the exon of the *OR4K15* gene (*Figure 6A*). The other three homologs of OR4K15-rbz belong to the same olfactory receptor family 4 subfamily K (*Figure 6C*). However, there was no exact match for LINE-1-rbz (*Figure 6A*). Interestingly, a total of 1154 LINE-1-rbz homologs were mapped to the LINE-1 retrotransposon according to the RepeatMasker (http://www.repeatmasker.org) annotation. *Figure 6B* shows the distribution of LINE-1-rbz homologs in different LINE-1 subfamilies in the human genome. Only three subfamilies L1PA7, L1PA8, and L1P3 (L1PA7-9) can be considered as abundant with LINE-1-rbz homologs (>100 homologs per family). The consensus sequences of all homologs obtained are shown in *Figure 6D*. To investigate the self-cleavage activity of these homologs, we mainly focused on the mismatches in the more conserved internal loops. The major differences between the five consensus sequences are the mismatches in the first internal loop. The widespread A12C substitution can be found in the majority of LINE-1-rbz homologs, this substitution leads to a one-base pair extension of the second stem (P2) but almost no activity (RÃ': 0.03) based on our deep mutational scanning result. Then we selected three homologs without A12C substitution for LINE-1-rbz for in vitro cleavage assay (*Figure 6E*). However we did not observe significant cleavage activity, this might be caused by GU substitutions in the stem region. For three homologs of OR4K15-rbz, we only found one homolog of OR4K15 with pronounced self-cleavage activity (*Figure 6F*). In addition, we performed a similar bioinformatic search of the TS-like ribozymes in other primate genomes. Similarly, the majority (15 out of 18) of primate genomes have a large number of LINE-1 homologs (>500) and the remaining three have essentially none. However, there was no exact match. Only one homolog has a single mutation (U38C) in the genome assembly of Gibbon (*Figure 6—figure supplement 1*). The majority of these homologs have three or more mismatches (*Figure 6—figure supplement 1*). For OR4K15-rbz, all representative primate genomes contain at least one exact match of the OR4K15-rbz sequence.

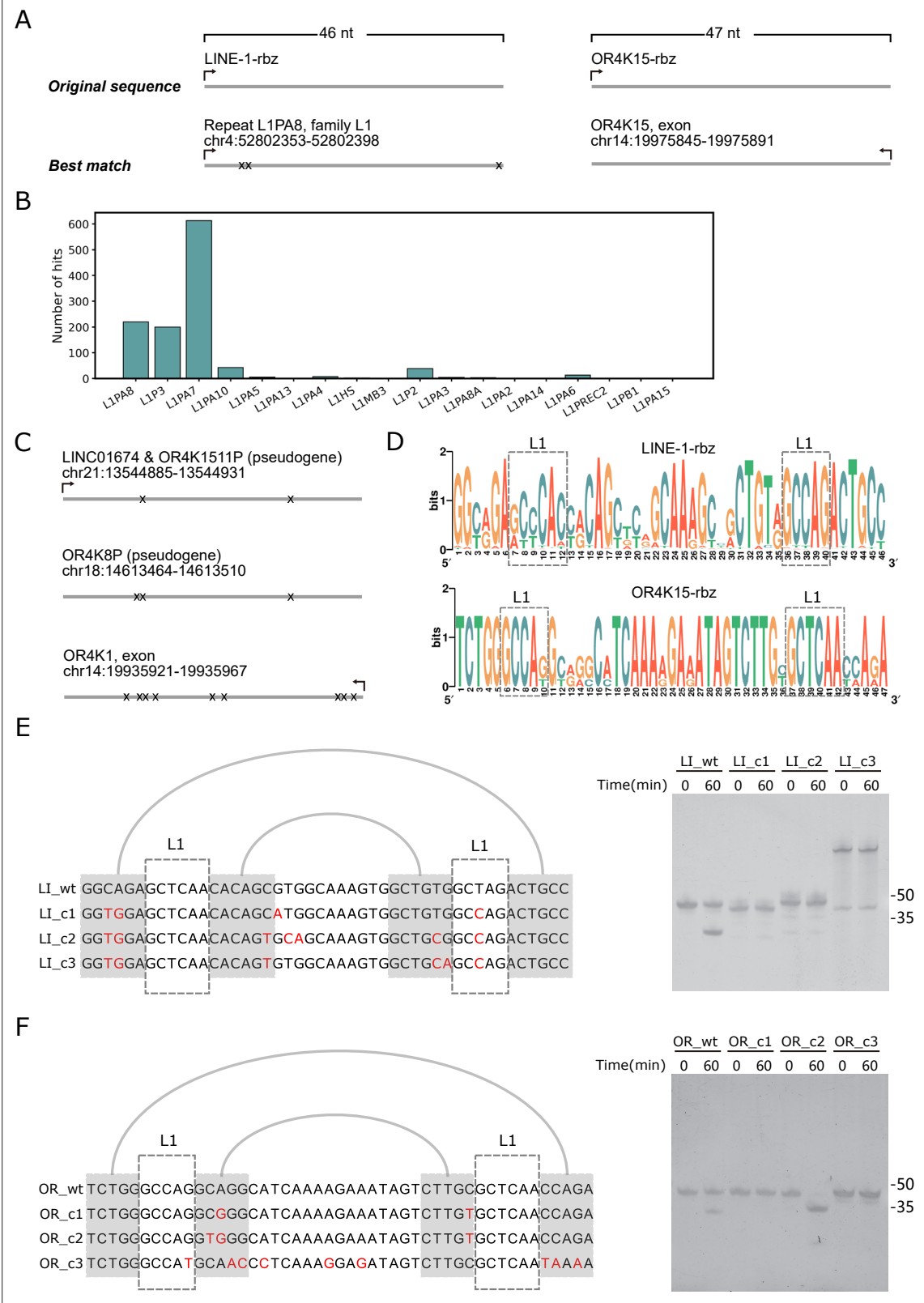

**Figure 6.** Homologous sequences of the two twister sister-like (TS-like) ribozymes. (**A**) Genomic locations of the sequences with the highest similarity to the LINE-1-rbz (left) and OR4K15-rbz (right). (**B**) The distribution of LINE-1-rbz homologs in different LINE-1 subfamilies of the human genome assembly (hg38). (**C**) Genomic locations of OR4K15-rbz homologs. (**D**) Nucleotide compositions of LINE-1-rbz homologs found in the human genome.

*Figure 6 continued on next page*

*Figure 6 continued*

(**E**) Sequence alignment (left) and PAGE result (right) of four LINE-1-rbz homologs. (**F**) Sequence alignment (left) and PAGE result (right) of four OR4K15-rbz homologs. OR_c1, OR_c2, and OR_c3 correspond to the three homologs (from top to bottom) shown in (**C**).

The online version of this article includes the following source data and figure supplement(s) for figure 6:

**Source data 1.** Original files of the full raw unedited gels displayed in *Figure 6*.

**Source data 2.** Figures with the uncropped gels displayed in *Figure 6*, with the relevant bands labelled.

**Figure supplement 1.** Secondary structure-based search of LINE-1-rbz and OR4K15-rbz in primate genomes.

## Discussion

The wide distribution of RNA self-cleaving activity in different species suggests that self-cleaving ribozymes play an important role in living organisms. Most self-cleaving ribozymes (*Deng et al., 2023*) are in non-coding regions of the hosting genome. For example, the CPEB3 ribozyme in the intron of the *CPEB3* gene (*Salehi-Ashtiani et al., 2006*), hammerhead ribozyme in the 3′ UTR of *Clec2* genes (*Martick et al., 2008*), hammerhead and HDV motifs associated with retrotransposons (*Eickbush and Eickbush, 2015*; *Sánchez-Luque et al., 2011*). LINE-1 belongs to the family of retrotransposons, they make up about 17% of the human genome, with over 500,000 copies (*Brouha et al., 2003*; *Lander et al., 2001*). The retrotransposition process of LINE-1 increases the copy number of repeats and generates insertional mutations in the genome, which contributes to genetic novelties as well as genomic instability. To date the known examples of self-cleaving ribozymes in retrotransposons are the aforementioned hammerhead, HDV-like and twister ribozymes. Especially, the HDV-like ribozymes have been found inside the 5′ UTR regions of the autonomous retrotransposons, including rDNA-specific (R2, R4, and R6) elements, RTEs (retrotransposon-like elements), telomere-specific SART, Baggins, L1Tc, etc. (*Ruminski et al., 2011*; *Weinberg et al., 2019*). Interestingly, those retrotransposon-related self-cleaving ribozymes have not been observed in mammalian genomes except for the Long Interspersed Nuclear Element-1 (LINE-1) ribozyme discussed in this study. However, we found that even this LINE-1-rbz sequence has at least two mismatches with the ancient human LINE-1 families, which led to the loss of self-cleavage activity. Similarly, most TS-like ribozyme homologs found in the primate LINE-1 retrotransposons seem to be inactive due to mutations, except for one homolog (U38C) found in Gibbon. Considering the location of the LINE-1 ribozyme inside the 5′ UTR, it suggests that this ribozyme might be related to the transcriptional regulation of the LINE-1 retrotransposon during evolution, like the HDV-like ribozymes. Since the majority of the LINE-1 retrotransposons inside the genome are 'DNA fossils' (*Beck et al., 2010*; *Lander et al., 2001*), which means that they were active in a specific period but became extinct (dysfunctional) during evolution. This suggests that the self-cleavage activity of the TS-like ribozymes in the LINE-1 retrotransposons may share a common evolutionary path with retrotransposition activity. By comparison, the OR4K15 ribozyme is located at the reverse strand of the coding region of the olfactory receptor gene *OR4K15*, suggesting a potentially novel functional mechanism for this ribozyme as antisense transcription is ubiquitous in mammals and may affect the gene regulation (*Vivancos et al., 2010*). However, the progress in understanding the functional roles of LINE-1 and OR4K15 self-cleaving ribozymes was slow because of lacking structural clues.

The centerpiece of structural clues is the RNA base-pairing structure resulted from the complex interplay of secondary and tertiary interactions. Previously, we developed a technique (*Zhang et al., 2020*) for inferring the base-pairing information of the self-cleaving ribozymes by deep mutational scanning. In this paper, we applied this technique to the LINE-1 and OR4K15 ribozymes, to obtain the functional regions and the base-pairing information. As both LINE-1 and OR4K15 ribozymes were discovered from the randomly fragmented human genomic DNA selection-based biochemical experiments, the original full-length (wild-type) sequences of the ribozymes do not necessarily represent the functional region of the RNAs. However, the functional regions could cleave when they were transcribed along with the upstream and downstream sequences, and, thus, they are more likely to be active in the cellular environment.

Indeed, we found that the functional regions for both ribozymes are very short (35 and 31 nucleotides for LINE-1 and OR4K15, respectively) and surprisingly in the same class in a circular-permutated form. We are confident about the secondary structures obtained because of the consistency between the results from the deep mutational study of the original sequence and the functional region of

the LINE-1 ribozyme. The obtained secondary structure is further confirmed by the discovery of the functional regions of the two TS-like ribozymes in the circular-permutated form from one another (*Figure 3A*). From the mutational information of LINE-1-rbz, the only mismatch (U38C) from OR4K15 in the catalytic region has an RA' of 0.6. While the corresponding mismatch mutation C8U in OR4K15 has an RA of 1.54, or 0.65 for U8C. This consistency in activity reduction for the U→C mutation further confirmed that these two ribozymes belong to the same class in a circular-permutated form. Currently, three classes of self-cleaving ribozymes, the hammerhead (*de la Peña and García-Robles, 2010b*; *Jimenez et al., 2011*; *Przybilski et al., 2005*; *Seehafer et al., 2011*), twister (*Roth et al., 2014*), and the hairpin (*Weinberg et al., 2021*) ribozymes are found in different species in multiple circular-permutated forms. The discovery of permutated TS-like ribozymes in the human genome for the first time suggests that we can find additional examples of circular-permutated TS-like ribozymes in other species.

More interestingly, we found that the internal loop regions of the TS-like ribozymes share a high similarity with the known catalytic region of the twister sister ribozyme (two mismatches only, *Figure 3A*). The conserved cleavage sites suggest a similar structure and mechanism as shown in *Figure 3A*. However, the maximum observed rate constant ($k_{obs}$) for the twister sister ribozyme was ~5 min$^{-1}$ (*Weinberg et al., 2015*), compared to ~0.11 ± 0.01 min$^{-1}$ for the TS-like ribozymes. This rate difference is not mainly caused by mismatches in the catalytic core between the twister sister and LINE-1/OR4K15 ribozymes. In fact, deep mutational scanning results indicate that one mismatch (U38C) in the catalytic region of OR4K15-rbz (*Figure 3A*) has an RA' of 0.6. Two mismatches (A12U and U38A) in the catalytic region of the twister sister ribozyme (*Figure 3A*) have the RA' of 0.28 and 0.47, respectively, whereas the coexistence of these two mismatches only has the RA' of 0.09. Thus, a more sophisticated structure along with long-range interactions involving the SL4 region in the twister sister ribozyme must have helped to stabilize the catalytic region for improved catalytic activity. Similarly, previous studies have demonstrated that peripheral regions of hammerhead (*De la Peña et al., 2003*), hairpin (*Zhao et al., 2000*), and HDV (*Tinsley and Walter, 2007*; *Webb et al., 2016*) ribozymes could greatly increase their self-cleavage activity. Given the importance of the peripheral regions, the absence of this tertiary interaction in the TS-like ribozyme may not be able to fully stabilize the structural form generated from homology modeling. Moreover, it is important to note that the catalytic regions of our TS-like ribozymes and the env-33 sequence from the multiple sequence alignment of twister sister ribozyme in a previous study (*Weinberg et al., 2015*) are quite identical (*Figure 3—figure supplement 5*), with only one mismatch in OR4K15-rbz and no mismatch in LINE-1-rbz. This high similarity further indicates that the TS-like ribozymes identified in this study should share the same origin as the twister sister ribozymes.

Indeed, we found some differences between the responses to mutations in the catalytic regions around the cleavage sites of TS-like ribozymes and those of twister sister ribozymes. Although there are pronounced differences around the cleavage sites in previous twister sister ribozyme structures, both cytosines at the cleavage sites were found to be insensitive to mutations (*Liu et al., 2017*; *Zheng et al., 2017*). However, unlike twister sister ribozymes, the cytosine at the cleavage site of TS-like ribozyme was sensitive to mutations in our mutation assays. According to our modeled structures, the H-bonding interactions associated with the cleavage site C10-A11 (C40-A41 in OR4K15-core) in TS-like ribozymes show that the cytosine (C10 in LINE-1-core, C40 in OR4K15) is relatively conserved, which is consistent with the mutation assay results. In the modeled structure, we did not observe direct contacts that helped to stabilize the scissile phosphate and the in-line alignments required for the catalytic step. This may suggest that conformational changes may be required to generate the active conformation prior to the catalytic step as in the twister sister ribozyme structures. Or this is due to inexact structures generated from homology modeling, because Mg$^{2+}$ was not included during homology modeling. Our experimental results (*Figure 5B–D*) showed that Mg$^{2+}$ is essential for both structure folding and catalysis. The absence of Mg$^{2+}$ during modeling might lead to the conformational change of the tertiary structures. Thus, further studies are needed to illustrate the possible structural and mechanistic differences between TS-like ribozymes and twister sister ribozymes.

According to the bioinformatic analysis result, there are some TS-like ribozymes (one LINE-1-rbz homolog in the Gibbon genome, and some OR4K15-rbz homologs) with in vitro cleavage activity in primate genomes. Unlike the more conserved CPEB3 ribozyme which has a clear function, the function of the TS-like ribozymes is not clear, as they are not conserved, belong to the pseudogene

or are located at the reverse strand. While the in vivo activities and functions of these two circular-permutated TS-like ribozymes require further studies, their discovery may have important therapeutics and biotech implications. First, the TS-like ribozymes have the simplest secondary structure with two stems and two internal loops, compared to all naturally occurring self-cleaving ribozymes. Previous studies also have revealed some minimized forms of self-cleaving ribozymes, including hammerhead (*Epstein and Gall, 1987*; *Lünse et al., 2017*) and HDV-like (*Riccitelli et al., 2014*) ribozymes. However, when comparing the conserved segments, they (≥36 nt) are not as short as the TS-like ribozymes (31 nt) found here. Second, this TS-like ribozyme after removing the stem-loop region only requires a 15-nucleotide enzyme strand and a 16-nucleotide substrate strand for its function. Thus, it can be easily modified into a trans-cleaving ribozyme, which was found useful for bioengineering and RNA-targeting therapeutics (*Huang et al., 2019*). A recent intracellular selection-based study has shown that hammerhead ribozyme can be engineered into a trans-acting ribozyme with consistent gene knockdown ability on different targeted mRNA either in prokaryotic or eukaryotic cells (*Huang et al., 2019*). The TS-like ribozyme, the simplest one known, may have an unprecedented advantage, compared to existing trans-cleaving ribozymes.

What is also important is that this work further confirms the usefulness of deep mutational scanning and high-throughput sequencing for RNA structural inference by CODA analysis (*Kobori and Yokobayashi, 2016*; *Zhang et al., 2020*). The method is not limited to self-cleaving ribozymes, if combined with other functional or phenotypic selection techniques. Moreover, the structural and full mutational information provided by this method could be utilized to discover additional functional RNAs and thus help us to investigate the hidden side of non-coding RNAs.

# Materials and methods

All the oligonucleotides listed in *Supplementary file 3*, except for the doped library, were purchased from IDT (Integrated DNA technologies) and Genscript. The doped mutant library Rz_LINE-1_doped with a doping rate of 6%, was purchased from the Keck Oligo Synthesis Resource at Yale University. All the high-throughput sequencing experiments were performed on the Illumina HiSeq X platform by Novogene Technology Co., Ltd.

## Deep mutational scanning experiments on original ribozyme sequences

The workflow for two deep mutational scanning experiments is illustrated in *Figure 1—figure supplement 6*. The deep mutational scanning of the original sequences of LINE-1 ribozyme (labeled as LINE-1-ori) and OR4K15 ribozyme (labeled as OR4K15-ori) followed the same protocol as our previous work (*Zhang et al., 2020*). Briefly, the mutant library was generated from three rounds of error-prone PCR, barcoded using primer Bar F and Bar R (*Supplementary file 3*), and then diluted. After that, the transcribed RNA of the barcoded mutant library was reverse transcribed with RT m13f adp1 and template switching oligo TSO (*Supplementary file 3*). The total cDNA was used to generate the RNA-seq library with primer P5R1 adp1 and P7R2 adp2 (*Supplementary file 3*). The DNA-seq libraries were generated by amplifying the ribozyme mutant library with primer P5R1_m13f and P7R2_t7p (*Supplementary file 3*). Both DNA-seq and RNA-seq libraries were sequenced on an Illumina HiSeq X sequencer with 25% PhiX control by Novogene Technology Co., Ltd. We counted the number of reads of the cleaved and uncleaved portions of each variant by mapping their respective barcodes. The relative activity (RA) of each variant is calculated by using the equation $RA\left(var\right) = N_{cleaved}\left(var\right) N_{total}\left(wt\right) / N_{total}\left(var\right) N_{cleaved}\left(wt\right)$.

## CODA analysis

We used CODA (*Zhang et al., 2020*) to analyze the RA of LINE-1-ori and OR4K15-ori. For MC simulated annealing, we used the same weighting factor of 2 for both datasets as employed previously (*Zhang et al., 2020*).

## Deep mutational scanning and CODA analysis on the functional region of LINE-1 ribozyme

We identified the functional region of the LINE-1 ribozyme (labeled as LINE-1-rbz) by removing the terminal sequence positions whose mutations do not affect cleavage activities. Then, the second round

of deep mutational scanning experiment was applied to the LINE-1-rbz. Here, we used a doped mutation library instead of a library generated by error-prone PCR. The doped mutant library Rz_LINE-1_doped (*Supplementary file 3*) was amplified by using primer T7prom and M13F (*Supplementary file 3*). The PCR product was then gel-purified, quantified, and diluted. Approximately $5 \times 10^5$ doped DNA molecules were amplified by T7prom and M13F primers (*Supplementary file 3*) to produce enough DNA templates for in vitro transcription. The transcribed and purified RNAs (10 pmol) of the mutant library were mixed with 100 pmol rM13R_5desBio_3P (*Supplementary file 3*), 2 µl 10× RtcB reaction buffer (New England Biolabs), 2 µl 1 mM GTP, 2 µl 10 mM MnCl₂, 0.2 µl of Murine RNase Inhibitor (40 U/µl, New England Biolabs), and 1 µl RtcB RNA ligase (New England Biolabs) to a total volume of 20 µl. The reaction mixture was incubated at 37°C for 1 hr, and then purified using RNA Clean & Concentrator-5 kit (Zymo Research). The purified products were mixed with 2 µl of 10 µl M RT_m13f_adp1 (*Supplementary file 3*) and 1 µl of 10 mM dNTPs in a volume of 8 µl, and then heated to 65°C for 5 min and placed on ice. Reverse transcription was initiated by adding 4 µl of 5× ProtoScript II Buffer (New England Biolabs), 2 µl of 0.1 M dithiothreitol (DTT), 0.2 µl of Murine RNase Inhibitor (40 U/µl, New England Biolabs) and 1 µl ProtoScript II RT (200 U/µl, New England Biolabs) to a total volume of 20 µl. The reaction mixture was incubated at 42°C for 1 hr, and then purified by Sera-Mag Magnetic Streptavidin-coated particles (Thermo Scientific).

The purified products after streptavidin-based selection were amplified by PCR to construct the RNA-seq library with primer P7R2_m13r and P5R1_m13f (*Supplementary file 3*). The DNA template for transcription was used as the template for constructing the DNA-seq library with primer P5R1_m13f and P7R2_t7p (*Supplementary file 3*). The DNA-seq and RNA-seq libraries were also sequenced on an Illumina HiSeq X sequencer with 25% PhiX control by Novogene Technology Co., Ltd. Raw paired-end sequencing reads were filtered and then merged by using the bioinformatic tool PEAR (*Zhang et al., 2014*) to generate the high-quality merged reads. We counted the read number of each variant in DNA-seq and RNA-seq by mapping the barcode. Then, we estimated the relative activity (RA′) of each variant by using the equation $RA'(var) = N_{RNAseq}(var) N_{DNAseq}(wt) / N_{DNAseq}(var) N_{RNAseq}(wt)$.

This relative activity (RA′) measure is different from the relative activity (RA) when both cleaved and uncleaved sequences were obtained in RNA-seq in the first-round deep mutational scanning. However, it should be a good estimate of RA because a mutant with a higher activity (cleavage rate) should have a higher possibility of being captured by RtcB ligase. We confirmed RA′ as a good estimate of RA by comparing shared single mutations from LINE-1-ori and LINE-1-rbz and obtained a strong Pearson correlation coefficient of 0.66 (Spearman correlation coefficient of 0.769). Moreover, both RA and RA′ show that the central region of LINE-1-rbz is insensitive to mutations (*Figure 1—figure supplement 5*). Moreover, secondary structures derived from RA′ and RA are consistent with each other (see below), confirming the correct identification of structural and functional regions of LINE-1 ribozyme.

We used the same method CODA (*Zhang et al., 2020*) to analyze the RA′ of LINE-1-rbz. For MC simulated annealing, we used the same weighting factor of 2.

## PAGE-based cleavage assays

Both ribozyme sequences were separated into two portions: the substrate part (LI_S_5F3T and OR_S_5F3T, *Supplementary file 3*) and the enzymatic part (LI_E and OR_E, *Supplementary file 3*). LI_S_5F3T and OR_S_5F3T were synthesized and HPLC purified by Genscript. Cytosine C10 and C40 were substituted by deoxycytosine in LI_S_dC10 and OR_S_dC40 (*Supplementary file 3*) during synthesis.

In a 20-µl reaction system, 40 pmol LI_S_5F3T or OR_S_5F3T was mixed with 2 µl 0.2 M Tris–HCl and 2 µl 10 mM metal ion stock solution unless noted otherwise. 60 pmol LI_E/OR_E (*Supplementary file 3*) was added to initiate the reaction. Reactions were incubated at 37°C for 1 hr, and then stopped by adding an equal volume of stop solution (95% formamide, 0.02% sodium dodecyl sulfate, 0.02% bromophenol blue, 0.01% Xylene Cyanol, 1 mM ethylenediaminetetraacetic acid). The reaction products were then separated by denaturing (8 M urea) 15% PAGE and visualized by SYBR Gold (Thermo Fisher) staining.

For bimolecular mutants of LINE-1-rbz and OR4K15-rbz, we separated the ribozymes into the substrate part (5′ 6-FAM labeled) and the enzymatic part in a similar way. In a 20-µl reaction system, 40 pmol substrate RNA was mixed with 2 µl 0.2 M Tris–HCl and 2 µl 10 mM Mg²⁺ stock solution. 60 pmol

enzyme RNA was added to initiate the reaction. Reactions were incubated at 37°C for different time intervals, and then stopped by adding an equal volume of stop solution (95% formamide, 0.02% SDS, 0.02% bromophenol blue, 0.01% Xylene Cyanol, 1 mM EDTA). The reaction products were then separated by denaturing (8 M urea) 15% PAGE.

For single-stranded mutants of LINE-1-rbz and OR4K15-rbz, we incubated 60 pmol RNA in a 20 μl reaction system. $Mg^{2+}$ stock solution was added to initiate the reaction. The final concentrations for the reaction solution were 20 mM Tris–HCl buffer (pH 7.5), 100 mM KCl, and 10 mM $MgCl_2$. Reactions were incubated at 37°C for different time intervals, and then stopped by adding an equal volume of stop solution (90% formamide, 0.02% SDS, 0.02% bromophenol blue, 0.01% Xylene Cyanol, 50 mM EDTA). The reaction products were then separated by denaturing (8 M urea) 12% PAGE.

## Fluorescence-based kinetic analysis

The fluorescence-based kinetic analysis of the two TS-like ribozymes was similar to our previous work on the CPEB3 ribozyme. The substrate RNAs LI_S_5F3T and OR_S_5F3T were synthesized with 5′ FAM as the fluorophore and 3′ TAMRA as the quencher. Cleavage of the substrate part by the enzymatic part will relieve the quenching and therefore generate a fluorescence signal.

For one 100 μl reaction system, 100 pmol enzyme RNA was mixed with 20 pmol substrate RNA unless noted otherwise. In the ion-dependent assay, a final concentration of 20 mM Tris–HCl solution (pH 7.5) and 100 mM KCl was used. Different concentrations of $MgCl_2$ or $MnCl_2$ solutions were employed to initiate the reactions.

The fluorescence (Ex 488 nm/Em 520 nm) was measured at 37°C for 3 hr using BioTek Synergy H1, and then normalized by the control (no divalent ions) at $t = 0$. The first-order rate constant of ribozyme cleavage $k_{obs}$ was calculated in a similar way as previously described (*Zhang et al., 2020*). The kinetics data were fitted to $F = A - Be^{-k_{obs}t}$, where $F$ is the fluorescence intensity, $t$ is the time, $A$ is the fluorescence at completion, and $B$ is the amplitude of the observable phase. Each data point in *Figure 5—figure supplement 2* is the average of $k_{obs}$ of three independent reactions.

## 3D structure modeling of the TS-like ribozymes

We matched the base pairs of the TS-like ribozymes to the most similar twister sister ribozyme (PDB: 5Y87) by map_align (*Ovchinnikov et al., 2017*). Then we modeled the RNA structure by replacing, inserting, and deleting the nucleotides with household RNA modeling Python script. Finally, we use RNA-BRiQ (*Xiong et al., 2021*) developed by our group to optimize the structure model. The source code can be accessed at https://gitee.com/hongxu66/coda-map.

## Secondary structure-based search of LINE-1-rbz and OR4K15-rbz

All representative primate genome assemblies in fasta format and the corresponding RepeatMasker (http://www.repeatmasker.org) annotation files were downloaded from the UCSC genomic website (http://genome.ucsc.edu). We used cmbuild from Infernal (*Nawrocki et al., 2009*; *Nawrocki and Eddy, 2013*) to build the covariance model (*Eddy and Durbin, 1994*) of LINE-1-rbz and OR4K15-rbz with the wild-type sequence and predicted secondary structure from deep mutational scanning experiment as input. Afterwards, we used the covariance model to search against the latest release of different representative primate genome assemblies by using cmsearch from Infernal. The hits with an $E$-value <1.0 were then used for downstream analysis.

## Acknowledgements

This work was funded by China Postdoctoral Science Foundation (fund no. 2021M702295 to ZZ) and a grant from National Natural Science Foundation of China (Grant #22350710182) to YZ. This work was also supported in part by Australian Research Council DP180102060 and DP210101875 and by National Health and Medical Research Council (1121629) of Australia to YZ. We would like to acknowledge the use of the High-Performance Computing Cluster 'Gowonda' at Griffith University and the supercomputing facility at the Shenzhen Bay Laboratory to complete this research. This research/project has also been undertaken with the aid of the research cloud resources provided by the Queensland Cyber Infrastructure Foundation (QCIF) and Australian Research Data Commons (ARDC). We also gratefully acknowledge the support of NVIDIA Corporation with the donation of the Titan V GPU used for this research.

## Additional information

### Funding

| Funder | Grant reference number | Author |
|---|---|---|
| China Postdoctoral Science Foundation | 2021M702295 | Zhe Zhang |
| National Natural Science Foundation of China | 22350710182 | Yaoqi Zhou |
| Australian Research Council | DP180102060 | Yaoqi Zhou |
| National Health and Medical Research Council | 1121629 | Yaoqi Zhou |
| Australian Research Council | DP210101875 | Yaoqi Zhou |

The funders had no role in study design, data collection and interpretation, or the decision to submit the work for publication.

### Author contributions

Zhe Zhang, Conceptualization, Data curation, Formal analysis, Investigation, Visualization, Methodology, Writing – original draft, Writing - review and editing; Xu Hong, Software, Investigation, Methodology; Peng Xiong, Formal analysis, Investigation, Methodology; Junfeng Wang, Supervision, Writing – original draft; Yaoqi Zhou, Supervision, Funding acquisition, Writing – original draft, Project administration, Writing - review and editing; Jian Zhan, Conceptualization, Supervision, Writing – original draft, Project administration

### Author ORCIDs

Zhe Zhang ⓘ http://orcid.org/0000-0001-7783-7683
Yaoqi Zhou ⓘ https://orcid.org/0000-0002-9958-5699

Reviewer #1 (Public review): https://doi.org/10.7554/eLife.90254.3.sa1
Author response https://doi.org/10.7554/eLife.90254.3.sa2

## Additional files

### Supplementary files

- MDAR checklist
- Supplementary file 1. Summary of the deep sequencing results.
- Supplementary file 2. Pattern-based similarity search results against the Rfam database.
- Supplementary file 3. Oligonucleotides used in this study.

### Data availability

Illumina sequencing data for the LINE-1-ori, OR4K15-ori, and LINE-1-rbz were submitted to the NCBI Sequence Read Archive (SRA) under SRA accession number PRJNA662002 (https://www.ncbi.nlm.nih.gov/sra/PRJNA662002). The homology modeling structures of LINE-1-core and OR4K15-core are available in ModelArchive (https://modelarchive.org/doi/10.5452/ma-chp4j for LINE-1-core; https://modelarchive.org/doi/10.5452/ma-bqqf8 for OR4K15-core).The source code for structure modeling can be accessed at https://gitee.com/hongxu66/coda-map. The processed data can be downloaded in https://github.com/zh3zh/TS-like-ribozyme (copy archived at *Zhang, 2024*).

The following datasets were generated:

| Author(s) | Year | Dataset title | Dataset URL | Database and Identifier |
|---|---|---|---|---|
| Zhang Z, Zhan J, Zhou Y | 2021 | deep mutational scanning of LINE-1 ribozyme | https://www.ncbi.nlm.nih.gov/sra/PRJNA662002 | NCBI Sequence Read Archive, PRJNA662002 |
| Zhang Z, Hong X, Zhou Y, Zhan J | 2024 | Homology model of the human LINE-1-core ribozyme | https://doi.org/10.5452/ma-chp4j | ModelArchive, 10.5452/ma-chp4j |
| Zhang Z, Hong X, Zhou Y, Zhan J | 2024 | Homology model of the OR4K15-core ribozyme | https://doi.org/ | ModelArchive, 10.5452/ma-bqqf8 |

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
