## [Editor Report · eLife Assessment]

This **important** study uncovers a surprising link between two self-cleaving RNAs that belong to the same structural family. The evidence supporting the main conclusions is **convincing** and based on extensive biochemical and bioinformatic analysis. This research will be of broad interest to RNA molecular biologists and biochemists.

---

## [Referee Report · Reviewer #1 (Public review)]

Summary:

The overall analysis and discovery of the common motif is important and exciting. Very few human/primate ribozymes have been published and this manuscript presents a detailed analysis of two of them. The minimized domains appear to be some of the smallest known self-cleaving ribozymes.

Strengths:

The manuscript is rooted in deep mutational analysis of the human OR4K15 and LINE1 ribozymes and subsequently in modeling of their active site based on the closely-related core of the TS ribozyme. The experiments support the HTS findings and provide convincing evidence that the ribozymes are structurally related to the core of the TS ribozyme, which has not been found in primates prior to this work.

---

## [Author Response]

The following is the authors’ response to the original reviews.

**Public Reviews:**

**Reviewer #1 (Public Review):**
Summary:The overall analysis and discovery of the common motif are important and exciting. Very few human/primate ribozymes have been published and this manuscript presents a relatively detailed analysis of two of them. The minimized domains appear to be some of the smallest known self-cleaving ribozymes.Strengths:The manuscript is rooted in deep mutational analysis of the OR4K15 and LINE1 and subsequently in modeling of a huge active site based on the closely-related core of the TS ribozyme. The experiments support the HTS findings and provide convincing evidence that the ribozymes are structurally related to the core of the TS ribozyme, which has not been found in primates prior to this work.Weaknesses:(1) Given that these two ribozymes have not been described outside of a single figure in a Science Supplement, it is important to show their locations in the human genome, present their sequence and structure conservation among various species, particularly primates, and test and discuss the activity of variants found in non-human organisms. Furthermore, OR4K15 exists in three copies on three separate chromosomes in the human genome, with slight variations in the ribozyme sequence. All three of these variants should be tested experimentally and their activity should be presented. A similar analysis should be presented for the naturally-occurring variants of the LINE1 ribozyme. These data are a rich source for comparison with the deep mutagenesis presented here. Inserting a figure (1) that would show the genomic locations, directions, and conservation of these ribozymes and discussing them in light of this new presentation would greatly improve the manuscript. As for the biological roles of known self-cleaving ribozymes in humans, there is a bioRxiv manuscript on the role of the CPEB3 ribozyme in mammalian memory formation (doi.org/10.1101/2023.06.07.543953), and an analysis of the CPEB3 functional conservation throughout mammals (Bendixsen et al. MBE 2021). Furthermore, the authors missed two papers that presented the discovery of human hammerhead ribozymes that reside in introns (by de la PeÁ{plus minus}a and Breaker), which should also be cited. On the other hand, the Clec ribozyme was only found in rodents and not primates and is thus not a human ribozyme and should be noted as such.

We thank this Reviewer for his/her input and acknowledgment of this work. To improve the manuscript, we have included the genomic locations in Figure 1A, Figure 6A and Figure 6C. And we have tested the activity of representative variants found in the human genome and discussed the activity of the variants in other primates. All suggested publications are now properly cited.

Line 62-66: It has been shown that single nucleotide polymorphism (SNP) in CPEB3 ribozyme was associated with an enhanced self-cleavage activity along with a poorer episodic memory (14). Inhibition of the highly conserved CPEB3 ribozyme could strengthen hippocampal-dependent long-term memory (15, 16). However, little is known about the other human self-cleaving ribozymes.

Line 474-501: Homology search of two TS-like ribozymes. To locate close homologs of the two TS-like ribozymes, we performed cmsearch based on a covariance model (38) built on the sequence and secondary structural profiles. In the human genome, we got 1154 and 4 homolog sequences for LINE-1-rbz and OR4K15-rbz, respectively. For OR4K15-rbz, there was an exact match located at the reverse strand of the exon of OR4K15 gene (Figure 6A). The other 3 homologs of OR4K15-rbz belongs to the same olfactory receptor family 4 subfamily K (Figure 6C). However, there was no exact match for LINE-1-rbz (Figure 6A). Interestingly, a total of 1154 LINE-1-rbz homologs were mapped to the LINE-1 retrotransposon according to the RepeatMasker (http://www.repeatmasker.org) annotation. Figure 6B showed the distribution of LINE-1-rbz homologs in different LINE-1 subfamilies in the human genome. Only three subfamilies L1PA7, L1PA8 and L1P3 (L1PA7-9) can be considered as abundant with LINE-1-rbz homologs (>100 homologs per family). The consensus sequences of all homologs obtained are shown in Figure 6D. In order to investigate the self-cleavage activity of these homologs, we mainly focused on the mismatches in the more conserved internal loops. The major differences between the 5 consensus sequences are the mismatches in the first internal loop. The widespread A12C substitution can be found in majority of LINE-1-rbz homologs, this substitution leads to a one-base pair extension of the second stem (P2) but almost no activity (RA’: 0.03) based on our deep mutational scanning result. Then we selected 3 homologs without A12C substitution for LINE-1-rbz for in vitro cleavage assay (Figure 6E). But we didn’t observe significant cleavage activity, this might be caused by GU substitutions in the stem region. For 3 homologs of OR4K15-rbz, we only found one homolog of OR4K15 with pronounced self-cleavage activity (Figure 6F). In addition, we performed similar bioinformatic search of the TS-like ribozymes in other primate genomes. Similarly, the majority (15 out of 18) of primate genomes have a large number of LINE-1 homologs (>500) and the remaining three have essentially none. However, there was no exact match. Only one homolog has a single mutation (U38C) in the genome assembly of Gibbon (Figure S15). The majority of these homologs have 3 or more mismatches (Figure S15). For OR4K15-rbz, all representative primate genomes contain at least one exact match of the OR4K15-rbz sequence.

Line 598-602: According to the bioinformatic analysis result, there are some TS-like ribozymes (one LINE-1-rbz homolog in the Gibbon genome, and some OR4K15-rbz homologs) with in vitro cleavage activity in primate genomes. Unlike the more conserved CPEB3 ribozyme which has a clear function, the function of the TS-like ribozymes is not clear, as they are not conserved, belong to the pseudogene or located at the reverse strand.

(2) The authors present the story as a discovery of a new RNA catalytic motif. This is unfounded. As the authors point out, the catalytic domain is very similar to the Twister Sister (or "TS") ribozyme. In fact, there is no appreciable difference between these and TS ribozymes, except for the missing peripheral domains. For example, the env33 sequence in the Weinberg et al. 2015 NCB paper shows the same sequences in the catalytic core as the LINE1 ribozyme, making the LINE1 ribozyme a TS-like ribozyme in every way, except for the missing peripheral domains. Thus these are not new ribozymes and should not have a new name. A more appropriate name should be TS-like or TS-min ribozymes. Renaming the ribozymes to lanterns is misleading.

Although we observed some differences in mutational effects, we agree with the reviewer that it is more appropriate to call them TS-like ribozymes. We have replaced all “lantern ribozyme” with “TS-like ribozyme” as suggested.

(3) In light of (2) the story should be refocused on the fact the authors discovered that the OR4K15 and LINE1 are both TS-like ribozymes. That is very exciting and is the real contribution of this work to the field.

We thank this Reviewer for their acknowledgement of this work. To improve the manuscript, we have re-named the ribozymes as suggested.

(4) Given the slow self-scission of the OR4K15 and LINE1 ribozymes, the discussion of the minimal domains should be focused on the role of peripheral domains in full-length TS ribozymes. Peripheral domains have been shown to greatly speed up hammerhead, HDV, and hairpin ribozymes. This is an opportunity to show that the TS ribozymes can do the same and the authors should discuss the contribution of peripheral domains to the ribozyme structure and activity. There is extensive literature on the contribution of a tertiary contact on the speed of self-scission in hammerhead ribozymes, in hairpin ribozyme it's centered on the 4-way junction vs 2-way junction structure, and in HDVs the contribution is through the stability of the J1/2 region, where the stability of the peripheral domain can be directly translated to the catalytic enhancement of the ribozymes.

We appreciate your question and the valuable suggestions provided. We have included the citations and discussion about the peripheral domains in other ribozymes.

Line 570-576: Thus, a more sophisticated structure along with long-range interactions involving the SL4 region in the twister sister ribozyme must have helped to stabilize the catalytic region for the improved catalytic activity. Similarly, previous studies have demonstrated that peripheral regions of hammerhead (49), hairpin (50) and HDV (51, 52) ribozymes could greatly increase their self-cleavage activity. Given the importance of the peripheral regions, absence of this tertiary interaction in the TS-like ribozyme may not be able to fully stabilize the structural form generated from homology modelling.

(5) The argument that these are the smallest self-cleaving ribozymes is debatable. LÝ1/4nse et al (NAR 2017) found some very small hammerhead ribozymes that are smaller than those presented here, but the authors suggest only working as dimers. The human ribozymes described here should be analyzed for dimerization as well (e.g., by native gel analysis) particularly because the authors suggest that there are no peripheral domains that stabilize the fold. Furthermore, Riccitelli et al. (Biochemistry) minimized the HDV-like ribozymes and found some in metagenomic sequences that are about the same size as the ones presented here. Both of these papers should be cited and discussed.

We apologize for any confusion caused by our previous statement. To clarify, we highlighted “35 and 31 nucleotides only” because 46 and 47 nt contain the variable hairpin loops which are not important for the catalytic activity. By comparing the conserved segments, the TS-like ribozyme discussed in this paper is the shortest with the simplest secondary structure. And we have replaced the terms “smallest” and “shortest” with “simplest” in our manuscript. The title has been changed to “Minimal twister sister (TS)-like self-cleaving ribozymes in the human genome revealed by deep mutational scanning”. All the publications mentioned have been cited and discussed. Regarding possible dimerization, we did not find any evidence but would defer it to future detailed structural analysis to be sure.

Line 605-608: Previous studies also have revealed some minimized forms of self-cleaving ribozymes, including hammerhead (19, 53) and HDV-like (54) ribozymes. However, when comparing the conserved segments, they (>=36 nt) are not as short as the TS-like ribozymes (31 nt) found here.

(6) The authors present homology modeling of the OR4K15 and LINE1 ribozymes based on the crystal structures of the TS ribozymes. This is another point that supports the fact that these are not new ribozyme motifs. Furthermore, the homology model should be carefully discussed as a model and not a structure. In many places in the text and the supplement, the models are presented as real structures. The wording should be changed to carefully state that these are models based on sequence similarity to TS ribozymes. Fig 3 would benefit from showing the corresponding structures of the TS ribozymes.

We thank the reviewer for pointing these out and we have already fixed them. We have replaced all “lantern ribozyme” with “TS-like ribozyme” as suggested. The term “Modelled structures” were used for representing the homology model. And we have included the TS ribozyme structure in Fig 3.

**Reviewer #2 (Public Review):**
Summary:This manuscript applies a mutational scanning analysis to identify the secondary structure of two previously suggested self-cleaving ribozyme candidates in the human genome. Through this analysis, minimal structured and conserved regions with imminent importance for the ribozyme's activity are suggested and further biochemical evidence for cleavage activity are presented. Additionally, the study reveals a close resemblance of these human ribozyme candidates to the known self-cleaving ribozyme class of twister sister RNAs. Despite the high conservation of the catalytic core between these RNAs, it is suggested that the human ribozyme examples constitute a new ribozyme class. Evidence for this however is not conclusive.Strengths:The deep mutational scanning performed in this study allowed the elucidation of important regions within the proposed LINE-1 and OR4K15 ribozyme sequences. Part of the ribozyme sequences could be assigned a secondary structure supported by covariation and highly conserved nucleotides were uncovered. This enabled the identification of LINE-1 and OR4K15 core regions that are in essence identical to previously described twister sister self-cleaving RNAs.Weaknesses:I am skeptical of the claim that the described catalytic RNAs are indeed a new ribozyme class. The studied LINE-1 and OR4K15 ribozymes share striking features with the known twister sister ribozyme class (e.g. Figure 3A) and where there are differences they could be explained by having tested only a partial sequence of the full RNA motif. It appears plausible, that not the entire "functional region" was captured and experimentally assessed by the authors.

We thank this Reviewer for his/her input and acknowledgment of this work. Because a similar question was raised by reviewer 1, we decided to name the ribozymes as TS-like ribozymes. Regarding the entire regions, we conducted mutational scanning experiments at the beginning of this study. The relative activity distributions (Figure 1B, 1C) have shown that only parts of the sequence contributes to the self-cleavage activity. That is the reason why we decided to focus on the parts of the sequence afterwards.

They identify three twister sister ribozymes by pattern-based similarity searches using RNA-Bob. Also comparing the consensus sequence of the relevant region in twister sister and the two ribozymes in this paper underlines the striking similarity between these RNAs. Given that the authors only assessed partial sequences of LINE-1 and OR4K15, I find it highly plausible that further accessory sequences have been missed that would clearly reveal that "lantern ribozymes" actually belong to the twister sister ribozyme class. This is also the reason I do not find the modeled structural data and biochemical data results convincing, as the differences observed could always be due to some accessory sequences and parts of the ribozyme structure that are missing.

We appreciate the reviewer for raising this question. As we explained in the last question, we now called the ribozymes as TS-like ribozymes. We also emphasize that the relative activity data of the original sequences have indicated that the other part did not make any contribution to the activity of the ribozyme. The original sequences provided in the Science paper (Salehi-Ashtiani et al. Science 2006) were generated from biochemical selection of the genomic library. It did not investigate the contribution of each position to the self-cleavage activity.

Highly conserved nucleotides in the catalytic core, the need for direct contacts to divalent metal ions for catalysis, the preference of Mn2+ oder Mg2+ for cleavage, the plateau in observed rate constants at ~100mM Mg2+, are all characteristics that are identical between the proposed lantern ribozymes and the known twister sister class.The difference in cleavage speed between twister sister (~5 min-1) and proposed lantern ribozymes could be due to experimental set-up (true single-turnover kinetics?) or could be explained by testing LINE-1 or OR4K15 ribozymes without needed accessory sequences. In the case of the minimal hammerhead ribozyme, it has been previously observed that missing important tertiary contacts can lead to drastically reduced cleavage speeds.

We thank the reviewer for this question. We now called the ribozymes as TS-like ribozymes. As we explained in the last question, the relative activity data of the original sequences have proven that the other part did not make any contribution to the activity of the ribozyme. Moreover, we have tested different enzyme to substrate ratios to achieve single turn-over kinetics (Figure S13). The difference in cleavage speed should be related to the absence of peripheral regions which do not exist in the original sequences of the LINE-1 and OR4K15 ribozyme. We have included the publications and discussion about the peripheral domains in other ribozymes.

Line 458-463: The kobs of LINE-1-core was ~0.05 min-1 when measured in 10mM MgCl2 and 100mM KCl at pH 7.5 (Figure S13). Furthermore, the single-stranded ribozymes exhibited lower kobs (~0.03 min-1 for LINE-1-rbz) (Figure S14) when comparing with the bimolecular constructs. This confirms that the stem loop region SL2 does not contribute much to the cleavage activity of the TS-like ribozymes.

Line 570-576: Thus, a more sophisticated structure along with long-range interactions involving the SL4 region in the twister sister ribozyme must have helped to stabilize the catalytic region for the improved catalytic activity. Similarly, previous studies have demonstrated that peripheral regions of hammerhead (49), hairpin (50) and HDV (51, 52) ribozymes could greatly increase their self-cleavage activity. Given the importance of the peripheral regions, absence of this tertiary interaction in the TS-like ribozyme may not be able to fully stabilize the structural form generated from homology modelling.

**Reviewer 2: (Recommendations For The Authors):**
Major pointsIt would have made it easier to connect the comments to text passages if the submitted manuscript had page numbers or even line numbers.

We thank the reviewer for pointing this out and we have already fixed it.

In the introduction: "...using the same technique, we located the functional and base-pairing regions of..." The use of the adjective functional is imprecise. Base-paired regions are also important for the function, so what type of region is meant here? Conserved nucleotides?

We thank the reviewer for pointing this out. We were describing the regions which were essential for the ribozyme activity. And we have defined the use of “functional region” in introduction.

Line 95: we located the regions essential for the catalytic activities (the functional regions) of LINE-1 and OR4K15 ribozymes in their original sequences.

In their discussion, the authors mention the possible flaws in their 3D-modelling in the absence of Mg2+. Is it possible to include this divalent metal ion in the calculations?

We thank the reviewer for this question. Currently, BriQ (Xiong et al. Nature Communications 2021) we used for modeling doesn’t include divalent metal ion in modeling.

Xiong, Peng, Ruibo Wu, Jian Zhan, and Yaoqi Zhou. 2021. “Pairing a High-Resolution Statistical Potential with a Nucleobase-Centric Sampling Algorithm for Improving RNA Model Refinement.” Nature Communications 12: 2777. doi:10.1038/s41467-021-23100-4.

Abstract:It is claimed that ribozyme regions of 46 and 47 nt described in the manuscript resemble the shortest known self-cleaving ribozymes. This is not correct. In 1988, hammerhead ribozymes in newts were first discovered that are only 40 nt long.

We apologize for any confusion caused by our previous statement. To clarify, we highlighted “35 and 31 nucleotides only” as 46 and 47 nt contain the variable hairpin loops which are not important for the catalytic activity. By comparing the conserved segments, the TS-like ribozyme discussed in this paper is the shortest with the simplest secondary structure. And we have replaced the terms “smallest” and “shortest” with “simplest” in our manuscript. The title has been changed to “Minimal TS-like self-cleaving ribozyme revealed by deep mutational scanning”.

The term "functional region" is, to my knowledge, not a set term when discussing ribozymes. Does it refer to the catalytic core, the cleavage site, the acid and base involved in cleavage, or all, or something else? Therefore, the term should be (1) defined upon its first use in the manuscript and (2) probably not be used in the abstract to avoid confusion to the reader.

We apologize for any confusion caused by our previous statement. To clarify, we have changed the term “functional region” in abstract. And we have defined the use of “functional region” in introduction.

Line 34-37: We found that the regions essential for ribozyme activities are made of two short segments, with a total of 35 and 31 nucleotides only. The discovery makes them the simplest known self-cleaving ribozymes. Moreover, the essential regions are circular permutated with two nearly identical catalytic internal loops, supported by two stems of different lengths.

Line 95: we located the regions essential for the catalytic activities (the functional regions) of LINE-1 and OR4K15 ribozymes in their original sequences.

The choice of the term "non-functional loop" in the abstract is a bit unfortunate. The loop might not be important for promoting ribozyme catalysis by directly providing, e.g. the acid or base, but it has important structural functions in the natural RNA as part of a hairpin structure.

We thank the reviewer for pointing this out and we have re-phrased the sentences.

Line 33-34: We found that the regions essential for ribozyme activities are made of two short segments, with a total of 35 and 31 nucleotides only.

Line 283: Removing the peripheral loop regions (Figures 1B and 1C) allows us to recognize that the secondary structure of OR4K15-rbz is a circular permutated version of LINE-1-rbz.

Results:Please briefly explain CODA and MC analysis when first mentioned in the results Figure (1) The more detailed explanation of these terms for Figure 2 could be moved to this part of the results section (including explanations in the figure legend).

We thank the reviewer for pointing this out and we included a brief explanation.

Line 150-154: CODA employed Support Vector Regression (SVR) to establish an independent-mutation model and a naive Bayes classifier to separate bases paired from unpaired (26). Moreover, incorporating Monte-Carlo simulated annealing with an energy model and a CODA scoring term (CODA+MC) could further improve the coverage of the regions under-sampled by deep mutations.

Please indicate the source of the human genomic DNA. Is it a patient sample, what type of tissue, or is it an immortalized cell line? It is not stated in the methods I believe.

We thank the reviewer for pointing this out. According to the original Science paper (Salehi-Ashtiani et al. Science 2006), the human genomic DNA (isolated from whole blood) was purchased from Clontech (Cat. 6550-1). In our study, we directly employed the sequences provided in Figure S2 of the Science paper for gene synthesis. Thus, we think it is unnecessary to mention the source of genomic DNA in the methods section of our paper.

Please also refer to the methods section when the calculation of RA and RA' values is explained in the main text to avoid confusion.

We thank the reviewer for pointing this out and we have fixed it.

Line 207-208: Figure 2A shows the distribution of relative activity (RA’, measured in the second round of mutational scanning) (See Methods) of all single mutations

For OR4K15 it is stated that the deep mutational scanning only revealed two short regions as important. However, there is another region between approx. 124-131 nt and possibly even at positions 47 and 52 (to ~55), that could contribute to effective RNA cleavage, especially given the library design flaws (see below) and the lower mutational coverage for OR4K15. A possible correlation of the mutations in these regions is even visible in the CODA+MC analysis shown in Figure 1D on the left. Why are these regions ignored in ongoing experiments?

We thank the reviewer for this question. As shown in Table S1, although the double mutation coverage of OR4K15-ori was low (16.2 %), we got 97.6 % coverage of single mutations. The relative activity of these single mutations was enough to identify the conserved regions in this ribozyme. Mutations at the positions mentioned by the reviewer did not lead to large reductions in relative activity. Since the relative activity of the original sequence is 1, we presumed that only positions with average relative activity much lower than 1 might contribute to effective cleavage.

Regarding the corresponding correlation of mutations in CODA+MC, they are considered as false positives generated from Monte Carlo simulated annealing (MC), because lack of support from the relative activity results.

Have the authors performed experiments with their "functional regions" in comparison to the full-length RNA or partial truncations of the full-length RNA that included, in the case of OR4K15, nt 47-131? Also for LINE-1 another stem region was mentioned (positions 14-18 with 30-34) and two additional base pairs. Were they included in experiments not shown as part of this manuscript?

We appreciate the reviewer for raising this question. We only compared the full-length or partial truncations of the LINE-1 ribozyme. Since the secondary structure predicted from OR4K15-ori data was almost the same as LINE-1, we didn’t perform deep mutagenesis on the partial truncation of the OR4K15. However, the secondary structure of OR4K15 was confirmed by further biochemical experiments.

Regarding the second question, the additional base pairs were generated by Monte Carlo simulated annealing (MC). They are considered as false positives because of low probabilities and lack of support from the deep mutational scanning results. The appearance of false positives is likely due to the imperfection of the experiment-based energy function employed in current MC simulated annealing.

Are there other examples in the literature, where error-prone PCR generates biases towards A/T nucleotides as observed here? Please cite!

We thank the reviewer for pointing this out and we have included the corresponding citation.

Line 161-162: The low mutation coverage for OR4K15-ori was due to the mutational bias (27, 28) of error-prone PCR (Supplementary Figures S1, S2, S3 and S4).

Line 170-171: whose covariations are difficult to capture by error-prone PCR because of mutational biases (27, 28).

The authors mention that their CODA analysis was based on the relative activities of 45,925 and 72,875 mutation variants. I cannot find these numbers in the supplementary tables. They are far fewer than the read numbers mentioned in Supplementary Table 2. How do these numbers (45,925 and 72,875) arise? Could the authors please briefly explain their selection process?

We apologize for any confusion caused by our previous statement. Our CODA analysis only utilized variants with no more than 3 mutations. The number listed in the supplementary tables is the total number of the variants. To clarify, we have included a brief explanation for these numbers.

Line 203-204: We performed the CODA analysis (26) based on the relative activities of 45,925 and 72,875 mutation variants (no more than 3 mutations) obtained for the original sequence and functional region of the LINE-1 ribozyme, respectively.

What are the reasons the authors assume their findings from LINE-1 can be used to directly infer the structure for OR4K15? (Third section in results, last paragraph)

We apologize for any confusion caused by our previous statement. We meant to say that the consistency between LINE-1-rbz and LINE-1-ori results suggested that our method for inferring ribozyme structure was reliable. Thus, we employed the same method to infer the structure of the functional region of OR4K15. To clarify, we have re-phrased the sentence.

Line 259-261: The consistent result between LINE-1-rbz and LINE-1-ori suggested that reliable ribozyme structures could be inferred by deep mutational scanning. This allowed us to use OR4K15-ori to directly infer the final inferred secondary structure for the functional region of OR4K15.

There are several occasions where the authors use the differences between the proposed lantern ribozymes and twister sister data as reasons to declare LINE-1 and OR4K15 a new ribozyme class. As mentioned previously, I am not convinced these differences in structure and biochemical results could not simply result from testing incomplete LINE-1 and OR4K15 sequences.

We apologize for any confusion caused by our previous statement. Despite we observed some differences in mutational effects, we agree with the reviewer that it is not convincing to claim them as a new ribozyme class. We have replaced all “lantern ribozyme” with “TS-like ribozyme” as the reviewer 1 suggested.

The authors state, that "the result confirmed that the stem loop SL2 region in LINE-1 and OR4K15 did not participate in the catalytic activity". To draw such a conclusion a kinetic comparison between a construct that contains SL2 and does not contain SL2 would be necessary. The given data does not suffice to come to this conclusion.

We appreciate the reviewer for raising this question. To address this, we performed gel-based kinetic analysis of these two ribozymes (Figure S14).

Line 458-462: The kobs of LINE-1-core under single-turnover condition was ~0.05 min-1 when measured in 10mM MgCl2 and 100mM KCl at pH 7.5 (Figure S13). Only a slightly lower value of kobs (~0.03 min-1) was observed for LINE-1-rbz (Figure S14). This confirms that the stem loop region SL2 does not contribute to the cleavage activity of the TS-like ribozymes.

Construct/Library design:The last 31 bp in the OR4K15 ribozyme template sequence are duplicated (Supplementary Table 4). Therefore, there are 2 M13 fwd binding sites and several possible primer annealing sites present in this template. This could explain the lower yield for the mutational analysis experiments. Did the authors observe double bands in their PCR and subsequent analysis? The experiments should probably be repeated with a template that does not contain this duplication. Alternatively, the authors should explain, why this template design was chosen for OR4K15.

We apologize for this mistake during writing. Our construct design for OR4K15 contains only one M13F binding site. We thank the reviewer for pointing this out and we have fixed the error.

Figure 5B: Where are the bands for the OR4K15 dC-substrate? They are not visible on the gel, so one has to assume there was no substrate added, although the legend indicates otherwise.Also this figure, please indicate here or in the methods section what kind of marker was used. In panels A and B, please label the marker lanes.

We apologize for this mistake and we have repeated the experiment. The marker lane was removed to avoid confusion caused by the inappropriate DNA marker.

The authors investigated ribozyme cleavage speeds by measuring the observed rate constants under single-turnover conditions. To achieve single-turnover conditions enzyme has to be used in excess over substrate. Usually, the ratios reported in the literature range between 20:1 (from the authors citation list e.g.: for twister sister (Roth et al 2014) and hatchet (Li et al. 2015)) or even ~100:1 (for pistol: Harris et al 2015, or others https://www.sciencedirect.com/science/article/pii/S0014579305002061). Can the authors please share their experimental evidence that only 5:1 excess of enzyme over the substrate as used in their experiments truly creates single-turnover conditions?

We greatly appreciate the Reviewer for raising this question. To address this, we performed kinetic analysis using different enzyme to substrate ratios (Figure S13). There is not too much difference in kobs, except that kobs reach the highest value of 0.048 min-1 when using 100:1 excess of enzyme over the substrate.

Line 458-460: The kobs of LINE-1-core under single-turnover condition was ~0.05 min-1 when measured in 10mM MgCl2 and 100mM KCl at pH 7.5 (Figure S13).

Citations:In the introduction citation number 12 (Roth et al 2014) is mentioned with the CPEB3 ribozyme introduction. This is the wrong citation. Please also insert citations for OR4K15 and IGF1R and LINE-1 ribozyme in this sentence.

We thank the reviewer for pointing this out and we now have fixed it.

Also in the introduction, a hammerhead ribozyme in the 3' UTR of Clec2 genes is mentioned and reference 16 (Cervera et al 2014) is given, I think it should be reference 9 (Martick et al 2008)

We thank the reviewer for pointing this out and we now have fixed it.

In the results section it is stated that, "original sequences were generated from a randomly fragmented human genomic DNA selection based biochemical experiment" citing reference 12. This is the wrong reference, as I could not find that Roth et al 2014 describe the use of such a technique. The same sentence occurs in the introduction almost verbatim (see also minor points).

We thank the reviewer for pointing this out and we now have fixed it.

Minor pointsHeadline:Either use caps for all nouns in the headline or write "self-cleaving ribozyme" uncapitalized

We thank the reviewer for pointing this out and we now have fixed it.

Abstract:1st sentence: in "the" human genome"Moreover, the above functional regions are..." - the word "above" could be deleted here"named as lantern for their shape"- it should be "its shape""in term of sequence and secondary structure"- "in terms""the nucleotides at the cleavage sites" - use singular, each ribozyme of this class has only one cleavage site

We thank the reviewer for pointing these out and we now have fixed them.

Introduction:Change to "to have dominated early life forms"Change to "found in the human genome"Please write species names in italics (*D. melanogaster*, B. mori)Please delete "hosting" from "...are in noncoding regions of the hosting genome"Please delete the sentence fragment/or turn it into a meaningful sentence: "Selection-based biochemical experiments (12).Change to "in terms of sequence and secondary structure, suggesting a more"Please reword the last sentence in the introduction to make clear what is referred to by "its", e.g. probably the homology model of lantern ribozyme generated from twister sister ribozymes?Please refer to the appropriate methods section when explaining the calculation of RA and RA'.

We thank the reviewer for pointing these out and we now have fixed them.

The last sentence of the second paragraph in the second section of the results states that the authors confirmed functional regions for LINE-1 and OR4K15, however, until that point the section only presents data on LINE-1. Therefore, OR4K15 should not be mentioned at the end of this paragraph.

In response to the reviewer's suggestions, we have removed OR4K15 from this paragraph.

Line 225-228: The consistency between base pairs inferred from deep mutational scanning of the original sequences and that of the identified functional regions confirmed the correct identification of functional regions for LINE-1 ribozyme.

Change to "Both ribozymes have two stems (P1, P2), to internal loops ..."

We thank the reviewer for pointing this out and we now have fixed it.

The section naming the "functional regions" of LINE-1 and OR4K15 lantern ribozymes should be moved after the section in which the circular permutation is shown and explained. Therefore, the headline of section three should read "Consensus sequence of LINE-1 and OR4K15 ribozymes" or something along these lines.

We thank the reviewer for pointing this out and we now have fixed it.

Line 308-309: Given the identical lantern-shaped regions of the LINE-1-rbz and OR4K15-rbz ribozyme, we named them twister sister-like (TS-like) ribozymes.

The statement on the difference between C8 in OR4K15 and U38 in LINE-1 should be further classified. As U38 is only 95% conserved. Is it a C in those other instances or do all other nucleotide possibilities occur? Is the high conservation in OR4K15 an "artifact" of the low mutation rate for this RNA in the deep mutational scanning?

We thank the reviewer for this question. Yes, the high conservation in OR4K15 an "artifact" of the low mutation rate for this RNA in the deep mutational scanning. That is why RA’ value is more appropriate to describe the conservation level of each position. We also mentioned this in the manuscript:

Line 287-288: The only mismatch U38C in L1 has the RA’ of 0.6, suggesting that the mismatch is not disruptive to the functional structure of the ribozyme.

Section five, first paragraph: instead of "two-stranded LINE-1 core" use the term "bimolecular", as it is more commonly used.

We thank the reviewer for pointing this out and we now have changed it.

Figure caption 3 headline states "Homology modelled 3D structure..."but it also shows the secondary structures of LINE1, OR4K15 and twister sister examples.

We thank the reviewer for pointing this out and we now have removed “3D”.

In Figure 3C, we see a nucleobase labeled G37, however in the secondary structure and sequence and 3D structural model there is a C37 at this position. Please correct the labeling.

We thank the reviewer for pointing this out and we now have fixed it.

Section 7 "To address the above question..." please just repeat the question you want to address to avoid any confusion to the reader.

We thank the reviewer for pointing these out and we have re-phrased this sentence.

Line 364: Considering the high similarity of the internal loops, we further investigated the mutational effects on the internal loop L1s.

Please rephrase the sentence "By comparison, mutations of C62 (...) at the cleavage site did not make a major change on the cleavage activity...", e.g. "did not lead to a major change" etc.Section 8, first paragraph: This result further confirms that the RNA cleavage in lantern...", please delete "further"Change to "analogous RNAs that lacked the 2' oxygen atom in the -1 nucleotide"MethodsChange to "We counted the number of reads of the cleaved and uncleaved..."Change to "...to produce enough DNA template for in vitro transcription."Change to "The DNA template used for transcription was used..." (delete while)

We thank the reviewer for pointing these out and we now have fixed them.

SupplementAll supplementary figures could use more detailed Figure legends. They should be self-explanatory.Fig S1/S2: how is "mutation rate" defined/calculated?

We thank the reviewer for pointing this out and we now have added a short explanation. The mutation rate was calculated as the proportion of mutations observed at each position for the DNA-seq library.

Fig S3/S4: axis label "fraction", fraction of what? How calculated?

We thank the reviewer for pointing this out and we now have added a short explanation. The Y axis “fraction” represents the ratio of each mutation type observed in all variants.

Fig S5: RA and RA' are mentioned in the main text and methods, but should be briefly explained again here, or it should be clearly referred to the methods. Also, the axis label could be read as average RA' divided by average RA. I assume that is not the case. I assume I am looking at RA' values for LINE-1 rbz and RA values for LINE-1-ori? Also, mention that only part of the full LINE-1-ori sequence is shown...

We thank the reviewer for pointing this out and we have now added a short explanation. The Y axis represents RA’ for LINE-1-rbz, or RA for LINE-1-ori. The part shown is the overlap region between LINE-1-rbz and LINE-1-ori. We apologize for any confusion caused by our previous statement.

Fig S9 the magenta for coloring of the scissile phosphate is hard to see and immediately make out.

We thank the reviewer for pointing this out and we now have added a label to the scissile phosphate.

Fig S10: Why do the authors only show one product band here? Instead of both cleavage fragments as in Figure 5?

We thank the reviewer for this question. We purposely used two fluorophores (5’ 6-FAM, 3’ TAMRA) to show the two product bands in Figure 5. In Fig S10, long-time incubation was used to distinguish catalysis based self-cleavage from RNA degradation. This figure was prepared before the purchasing of the substrate used in Figure 5. The substrate strand used in Fig S10 only have one fluorophore (5’ 6-FAM) modification. And the other product was too short to be visualized by SYBR Gold staining.

Fig S13: please indicate meaning of colors in the legend (what is pink, blue, grey etc.)Please change to "RtcB ligase was used to capture the 3' fragment after cleavage...."

We thank the reviewer for pointing this out and we now have fixed it.